# Mitochondrial protein C15ORF48 is a stress-independent inducer of autophagy that regulates oxidative stress and autoimmunity

Yuki Takakura[1,2,3], Moeka Machida[1,2], Natsumi Terada[1,2], Yuka Katsumi[1], Seika Kawamura[1], Kenta Horie [3], Maki Miyauchi[3,4], Tatsuya Ishikawa[3,4], Nobuko Akiyama[3], Takao Seki [3], Takahisa Miyao[3,4], Mio Hayama[3,4], Rin Endo[3,4], Hiroto Ishii[3,4], Yuya Maruyama[3,4], Naho Hagiwara[3], Tetsuya J. Kobayashi[5], Naoto Yamaguchi[2], Hiroyuki Takano[1], Taishin Akiyama [3,4] ✉ & Noritaka Yamaguchi [1,2,3] ✉

Autophagy is primarily activated by cellular stress, such as starvation or mitochondrial damage. However, stress-independent autophagy is activated by unclear mechanisms in several cell types, such as thymic epithelial cells (TECs). Here we report that the mitochondrial protein, C15ORF48, is a critical inducer of stress-independent autophagy. Mechanistically, C15ORF48 reduces the mitochondrial membrane potential and lowers intracellular ATP levels, thereby activating AMP-activated protein kinase and its downstream Unc-51-like kinase 1. Interestingly, C15ORF48-dependent induction of autophagy upregulates intracellular glutathione levels, promoting cell survival by reducing oxidative stress. Mice deficient in *C15orf48* show a reduction in stress-independent autophagy in TECs, but not in typical starvation-induced autophagy in skeletal muscles. Moreover, *C15orf48*[−/−] mice develop autoimmunity, which is consistent with the fact that the stress-independent autophagy in TECs is crucial for the thymic self-tolerance. These results suggest that C15ORF48 induces stress-independent autophagy, thereby regulating oxidative stress and self-tolerance.

Autophagy is a process in which cytoplasmic components of a cell are transported to lysosomes and degraded by multiple enzymes. Initiation of autophagy typically depends on cellular stress, such as starvation or mitochondrial damage. Starvation-induced autophagy is often referred to as macroautophagy, which facilitates recycling of key metabolites, such as nucleotides, amino acids, and lipids to support cell survival. Mitochondrial damage-induced autophagy, known as mitophagy, selectively degrades damaged mitochondria and protects cells from mitochondrial stress[1-4].

Under normal conditions, the mammalian target of rapamycin (mTOR) protein kinase complex represses autophagy by inactivating Unc-51-like kinase 1 (ULK1) by phosphorylating it at Ser757[5,6]. Starvation inhibits mTOR activity, in turn leading to activation of ULK1. ULK1 phosphorylates autophagy-related gene 14 (ATG14), Beclin1, and other autophagy proteins to initiate autophagy[7]. Mitochondrial stress causes a reduction in intracellular ATP that triggers activation of the AMP-activated protein kinase (AMPK) complex, which is involved in various events related to mitochondrial homeostasis. For instance, if the AMPK

[1]Department of Molecular Cardiovascular Pharmacology, Graduate School of Pharmaceutical Sciences, Chiba University, Chiba 260-8675, Japan. [2]Laboratory of Molecular Cell Biology, Graduate School of Pharmaceutical Sciences, Chiba University, Chiba 260-8675, Japan. [3]Laboratory for Immune Homeostasis, RIKEN Center for Integrative Medical Sciences, Yokohama 230-0045, Japan. [4]Immunobiology, Graduate School of Medical Life Science, Yokohama City University, Yokohama 230-0045, Japan. [5]Institute of Industrial Science, The University of Tokyo, Tokyo 153-8505, Japan. ✉e-mail: taishin.akiyama@riken.jp; yamaguchinoritaka@chiba-u.jp

complex activates ULK1 by phosphorylating Ser555[8], then, activated ULK1 initiates mitophagy by phosphorylating the key mitophagy-inducing ubiquitin ligase, Parkin[9].

In addition to the stress-dependent autophagy, autophagy is induced by some types of stimuli[10]. IL-6 signaling reportedly triggers macroautophagy in cancer cells[11], which may promote chemotherapy resistance in colorectal cancer. Moreover, autophagy is constitutively active in a few tissues such as thymic epithelial cells (TECs), lens epithelial cells, and podocytes, without starvation in mice[12]. Mechanisms underlying these stress-independent autophagy remain to be elucidated.

TECs are self-antigen-presenting cells and are required for differentiation and selection of major histocompatibility complex (MHC)-restricted and self-tolerant T cells[13,14]. TECs are separated into cortical TECs (cTECs) and medullary TECs (mTECs), based on their localization in the thymus, and both types of TECs display self-antigen peptide and MHC complexes (self-pMHC) on their cell surfaces. cTECs promote survival of thymocytes expressing T cell antigen receptor (TCR) with moderate affinity for self-pMHC. In contrast, mTECs uniquely express and present thousands of tissue specific self-antigens (TSAs) under regulation of autoimmune regulator (AIRE) and other transcriptional regulators[15–17], thereby eliminating T cells that recognize self-pMHC with high affinity or converting them to regulatory T cells (Tregs).

For cell surface expression of self-pMHC by TECs, degradation of cytoplasmic self-proteins is essential for loading self-peptides on surface MHC molecules. Whereas MHC class II normally presents extracellular antigens, autophagy would permit presentation of intracellular antigens on MHC class II through lysosomal pathways. Previous studies revealed that TECs show high autophagy activity without starvation and infections[12,18]. This implies that such stress-independent autophagy might contribute to self-protein degradation for generating self-antigen peptides in TECs. Consistently, some studies have suggested that autophagy in TECs is important for acquisition of T cell self-tolerance[18–21]. However, the mechanism by which TECs induce autophagy in the absence of starvation is unknown.

Chromosome 15 open reading frame 48 (C15ORF48) (also known as NMES1, Coxfa4l3, MISTRV, or MOCCI) is a small mitochondrial protein composed of 83 amino acids, which is thought to modulate cytochrome *c* oxidase in electron transport chain complex IV (CIV)[22–25]. Expression of C15ORF48 is induced by inflammatory stimuli, such as interleukin (IL)−1β, interferon-γ, toll-like receptor ligands, and viral infection[23,24,26]. Nuclear factor-κB (NF-κB) is a pivotal signaling pathway in *C15ORF48* expression[26]. C15ORF48 reduces CIV activity, mitochondrial membrane potential, and reactive oxygen species (ROS) production, and protects against cell death from viral infection[23,27]. The 3′ untranslated region (UTR) of *C15ORF48* encodes microRNA miR-147b, which targets the *C15ORF48* homolog, *NDUFA4*. Induction of C15ORF48 is accompanied by miR-147b expression, thereby repressing *NDUFA4* expression via miR-147b[23,24,26,27]. Previous studies suggested that C15ORF48 replaces NDUFA4 protein in CIV during inflammatory stimuli. This mechanism explains the reduction of CIV activity by expression of C15ORF48 and miR-147b. However, the mechanism underlying the reduction in mitochondrial activity and ROS by C15ORF48 remain unclear[23].

In this work, we show that C15ORF48 promotes autophagy independently of starvation or mitochondrial stress. Mechanistically, C15ORF48 reduces mitochondrial activity and intracellular ATP levels, thereby inducing AMPK-ULK1 signaling. Surprisingly, C15ORF48-induced autophagy increases glutathione levels and thereby protect cells from oxidative stress. *C15orf48*$^{-/-}$ mice show a severe reduction in constitutive autophagy in mTECs and exhibit autoimmunity. Taken together, these results strongly suggest that C15ORF48 is the factor responsible for initiation of stress-independent autophagy and that it regulates oxidative stress and self-tolerance.

## Results

### C15ORF48 expression leads to reduction of intracellular ATP and activation of the AMP-activating kinase complex

To specifically investigate the role of C15ORF48 protein, independent of miR-147b, we first established a cell line stably expressing *C15ORF48* lacking its 5′ or 3′ UTR using A549 human lung cancer cells. As previously reported[22–24,28], C15ORF48 predominantly localized in mitochondria (Fig. 1a). Forced expression of C15ORF48 reduced mitochondrial membrane potential and intracellular ATP levels (Fig. 1b, c), suggesting that C15ORF48 suppresses mitochondrial activity. A reduction in intracellular ATP levels causes activation of AMPK complex[29]. Consistently, we found that C15ORF48-expressing cells showed increased phosphorylation of AMPKα (Fig. 1d). Thus, these data suggested that increased expression of C15ORF48 reduces mitochondrial activity in the absence of miR-147b, thereby activating AMPK due to the reduction in intracellular ATP.

### C15ORF48 is an autophagy inducer, independent of starvation

AMPK activates various metabolic processes[29], including autophagy[7,8]. We hypothesized that C15ORF48 could be a natural autophagy inducer because C15ORF48 expression results in phosphorylation of ULK1, a downstream kinase of AMPK, critical for autophagy induction (Fig. 1d). To verify this hypothesis, autophagy induction was evaluated by detecting LC3-II levels and the number of LC3 puncta ascribed to autophagosome formation, hallmarks of autophagy[30,31]. Detection of autophagosome formation was enhanced by addition of Bafilomycin A1 (Baf A1), which allows sensitive detection of LC3-II accumulation and autophagosome formation by inhibiting fusion of autophagosomes and lysosomes. We found that forced expression of C15ORF48 increased the level of LC3-II and the number of LC3 puncta, even under non-starved conditions, i.e., 4 h after replacement of culture media (Fig. 2a−c). C15ORF48-induced autophagy was repressed by the ULK1 inhibitor, SBI-0206965 (Fig. 2d−f), supporting the involvement of AMPK-ULK1 signaling in C15ORF48-induced autophagy. In contrast, phosphorylated mTOR, a hallmark of starvation-dependent autophagy[32], was not reduced by forced expression of C15ORF48 (Fig. 2g), suggesting that starvation-dependent autophagy would not be induced. Thus, C15ORF48-induced autophagy is distinct from the starvation-stress pathway of autophagy.

Because C15ORF48 affected mitochondrial activity, C15ORF48-expressing cells might cause mitophagy, leading to selective degradation of mitochondria[33]. However, the number of mitophagy dye-positive cells did not change in C15ORF48-expressing cells under normal conditions (Fig. 2h). We further investigated the relationship between C15ORF48 and mitophagy using the mitochondrial uncoupler, CCCP (carbonyl cyanide *m*-chlorophenylhydrazone). While treatment with CCCP increased the number of mitophagy dye-positive cells, this increment was significantly repressed in C15ORF48-expressing cells (Supplementary Fig. 1a). CCCP treatment induced stabilization of phosphatase and tensin homolog (PTEN)-induced kinase 1 (PINK1), an initial step of mitophagy[34,35], and PINK1 stabilization was significantly suppressed by forced expression of C15ORF48 (Supplementary Fig. 1b). We also analyzed colocalization of LC3 puncta with mitochondria. Under the normal culture condition, colocalization of LC3 puncta with mitochondria was low and showed no apparent difference between control and C15ORF48-expressing cells, consistent with the data in Fig. 2h which suggest that C15ORF48 expression does not induce mitophagy. Remarkably, CCCP treatment elevated colocalization of LC3 puncta with mitochondria, suggesting the onset of mitophagy, and this elevation was significantly suppressed in C15ORF48-expressing cells (Supplementary Fig. 1c, d). These results indicate that C15ORF48 has a suppressive, rather than promotive, role in mitophagy. Accordingly, although increased expression of C15ORF48 caused a reduction of mitochondrial activity and AMPK activation, it was not sufficient to induce mitophagy. Overall, these

results suggest that C15ORF48 is an inducer of starvation-independent macroautophagy, that activates AMPK-ULK1 signaling axis, which is triggered by reduction of intracellular ATP levels due to mild repression of mitochondrial activity.

## C15ORF48 is required for autophagy induced by inflammatory signaling

To determine the role of C15ORF48-induced autophagy in physiological situations, we focused on inflammatory signaling. Inflammatory cytokines, such as IL-1α and tumor necrosis factor (TNF), induced expression of C15ORF48 (Supplementary Fig. 2a, b)[23,24,26]. These cytokines activate transcription factor NF-κB for expression of its responsive genes. The NF-κB component, RelA, binds to the promoter region of the *C15ORF48* gene, and the NF-κB inhibitor (SC-514) repressed its induction. Thus, NF-κB is necessary for and directly regulates *C15ORF48* mRNA expression (Supplementary Fig. 2c, d).

As expected, C15ORF48 protein up-regulated by IL-1α was predominantly localized in mitochondria (Fig. 3a). Moreover, IL-1α stimulation reduced mitochondrial membrane potential and intracellular ATP levels (Fig. 3b, c). Notably, as observed in forced expression of C15ORF48, IL-1α stimulation increased autophagy activity and activation of AMPK and ULK1, even under non-starved conditions (Fig. 3d–f). Importantly, these IL-1α-inducing responses were repressed by *C15ORF48* knockdown, demonstrating their dependency on C15ORF48 (Fig. 3b, c, g–i). Additionally, the ULK1 inhibitor repressed IL-1α-induced autophagy (Fig. 3j–l). These results suggest that inflammatory stimuli induce expression of C15ORF48 to promotes stress-independent autophagy via AMPK-ULK1 signaling.

## High expression of endogenous C15ORF48 increases autophagy

Autophagy promotes survival and metastasis of tumor cells[36,37]. Therefore, we analyzed highly invasive breast cancer cells, MDA-MB-231, showing constitutively high NF-κB activation[38], which might be involved in malignancies[39]. MDA-MB-231 cells displayed high expression of endogenous C15ORF48 compared with A549 cells, which have lower constitutive NF-κB activity (Fig. 4a). Notably, MDA-MB-231 cells exhibited higher basal autophagy activity than A549 cells (Fig. 4a–c). Furthermore, *C15ORF48* knockdown increased intracellular ATP levels and repressed the phosphorylation levels of AMPKα and ULK1 in MDA-MB-231 cells (Fig. 4d, e). *C15ORF48* knockdown also suppressed basal autophagy activity (Fig. 4e–g). These results suggest that high expression of endogenous C15ORF48 activates stress-independent autophagy in tumor cells.

## C15ORF48-induced autophagy increased intracellular glutathione levels to protect cells from oxidative stress

C15ORF48 reduces intracellular ROS levels. However, its mechanism does not appear related to the reduction of CIV activity[23]. We suspected that C15ORF48 expression could cause an incremental increase in levels of natural antioxidants, thereby reducing ROS levels. Surprisingly, we found that levels of total glutathione and reduced glutathione (GSH) were significantly increased by forced expression of C15ORF48. Importantly, glutathione induction was repressed by *ATG5*- or *ATG7*-knockdown, indicating that autophagy induced by C15ORF48 is responsible for increasing the glutathione level (Fig. 5a–c). Interestingly, forced expression of C15ORF48 protected cells from cell death and repressed caspase-3 activation under oxidative stress

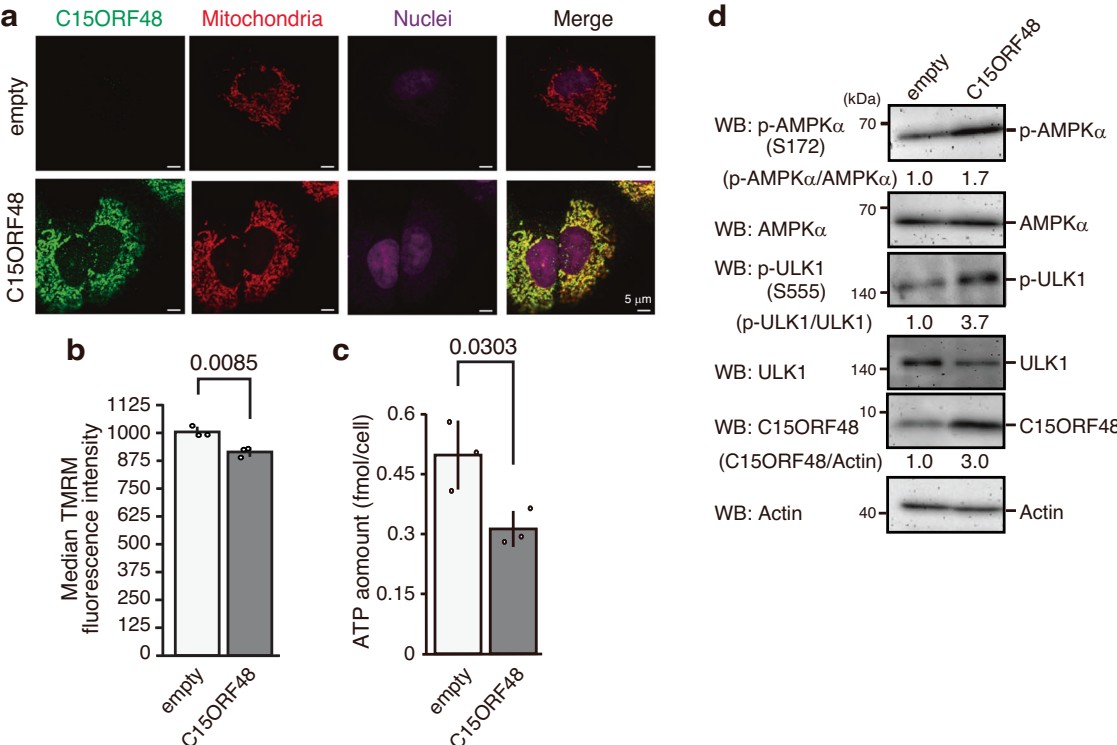

**Fig. 1 | C15ORF48 expression activates AMPK-ULK1 signaling. a** A549 cells stably expressing *C15ORF48* (A549/C15ORF48) or the empty (A549/empty) vector were fixed and stained with anti-C15ORF48 antibody and MitoTracker. Nuclei were counter-stained with TO-PRO-3. Representative images are shown from three independent experiments. **b** A549/empty and A549/C15ORF48 cells were analyzed for mitochondrial membrane potential by staining with TMRM (30 nM) for 30 min. Median TMRM fluorescence intensities are shown as the mean ± SD. Statistical significance was calculated using two-tailed unpaired Student's *t* test (*n* = 3,

biological replicates). **c** A549/empty and A549/C15ORF48 cells were analyzed for intracellular ATP levels. ATP amount in a cell is shown as the mean ± SD. Statistical significance was calculated using two-tailed unpaired Student's *t* test (*n* = 3, biological replicates). **d** A549/empty and A549/C15ORF48 cells were lysed and subjected to western blotting with indicated antibodies. Band intensity was measured, and quantitative ratios are shown. Data are representative of three independent experiments with three biological replicates.

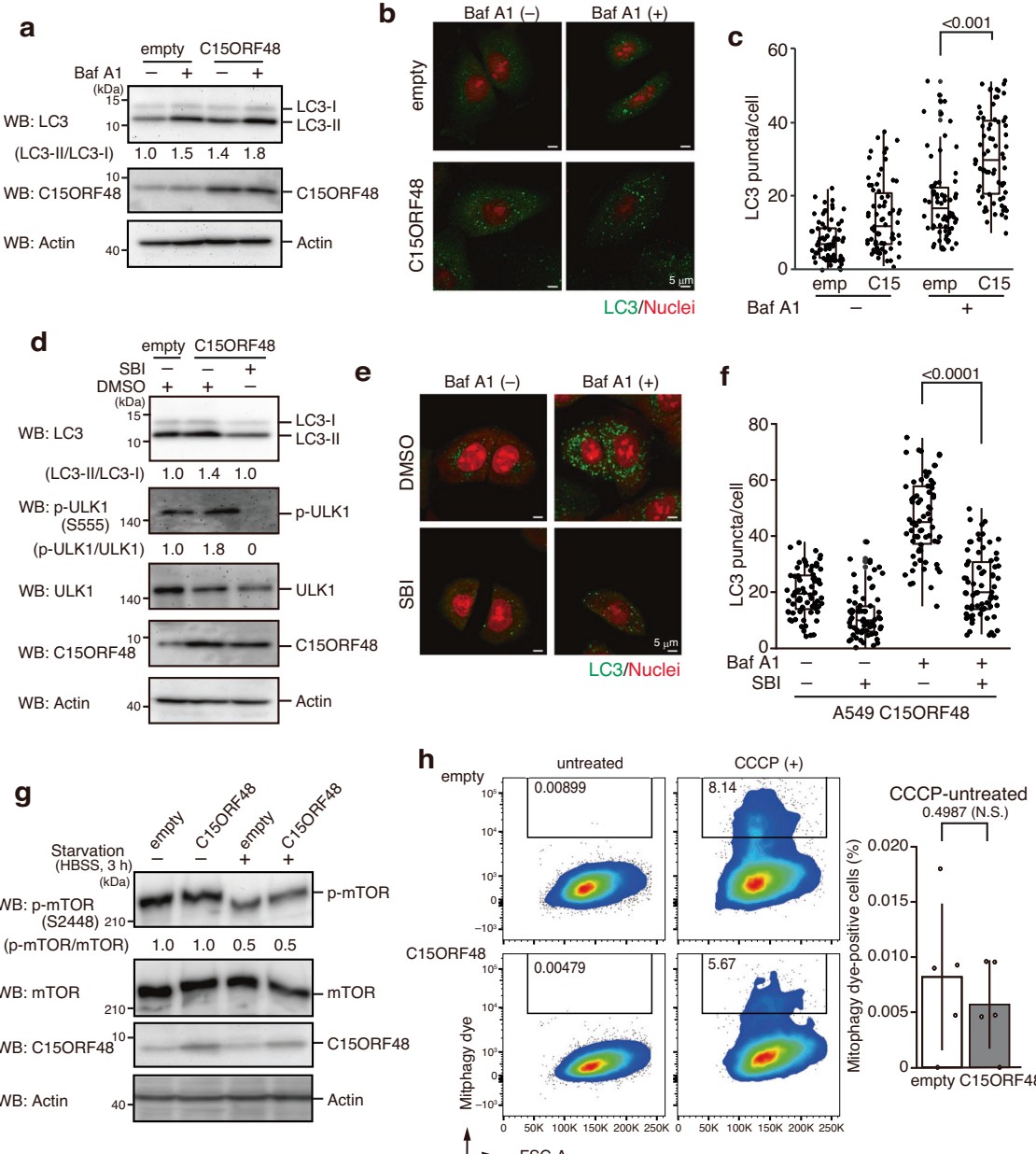

**Fig. 2 | C15ORF48 expression activates autophagy independently of starvation or mitochondrial stress. a** A549/empty and A549/C15ORF48 cells were treated with bafilomycin A1 (Baf A1, 200 μM, 1 h) or untreated 3 h after replacement of culture media and then subjected to western blotting as in Fig. 1d. **b, c** A549/empty and A549/C15ORF48 cells were treated with bafilomycin A1 (Baf A1, 200 μM, 1 h) or untreated 3 h after replacement of culture media. Cells were fixed and stained with anti-LC3 antibodies. Nuclei were counter-stained with PI (**b**). Numbers of LC3 puncta in each cell were calculated and are shown as means ± SDs. Statistical significance was calculated using two-way ANOVA followed by Tukey's multiple comparisons test ($n = 70$ cells from two independent experiments) (**c**). **d** A549/empty and A549/C15ORF48 cells were incubated for 4 h after replacement of culture media with fresh media containing DMSO (0.1%) or SBI-0206965 (SBI, 10 μM). After incubation, cells were subjected to western blotting as in Fig. 1d. **e, f** A549/C15ORF48 cells were treated with bafilomycin A1 (Baf A1, 200 μM, 1 h) or left

untreated 3 h after replacement of culture media with fresh media containing DMSO (0.1%) or SBI-0206965 (SBI, 10 μM). Cells were fixed and stained with anti-LC3 antibody. LC3 puncta and nuclei were visualized as in Fig. 2b (**e**). Numbers of LC3 puncta in each cell were calculated as shown in Fig. 2c. Statistical significance was calculated using two-way ANOVA followed by Tukey's multiple comparisons test ($n = 70$ cells from two independent experiments) (**f**). **g** A549/empty and A549/C15ORF48 cells were unstarved or starved with HBSS (3 h) 1 h after replacement of culture media and then subjected to western blotting as in Fig. 1d. **h** A549/empty and A549/C15ORF48 cells were treated with Mitophagy dye, and Mitophagy dye-positive cells (boxed areas) were detected by flow cytometry (left). Percentages of mitophagy dye-positive cells in CCCP-untreated cells, which show basal mitophagy levels, are shown as the mean ± SD (right). Statistical significance was calculated using two-tailed unpaired Student's $t$ test ($n = 5$, biological replicates). Not significant, N.S.

condition ($H_2O_2$ treatment) (Fig. 5d, e). In addition, C15ORF48-expressing cells showed increased susceptibility to the glutathione peroxidase (GPX) inhibitor, RSL3 (Fig. 5f). These data suggest that C15ORF48 induces up-regulation of glutathione levels, thereby reducing ROS levels and enhancing resistance to oxidative stress.

Consistently, IL-1α stimulation also increased levels of total glutathione and GSH, and showed enhanced cell viability to oxidative stress and susceptibility to RSL3 (Fig. 5g–i). Importantly, these changes were reversed by *C15ORF48* knockdown (Fig. 5g–i). Moreover, changes in glutathione levels and cell susceptibility to oxidative stress were also

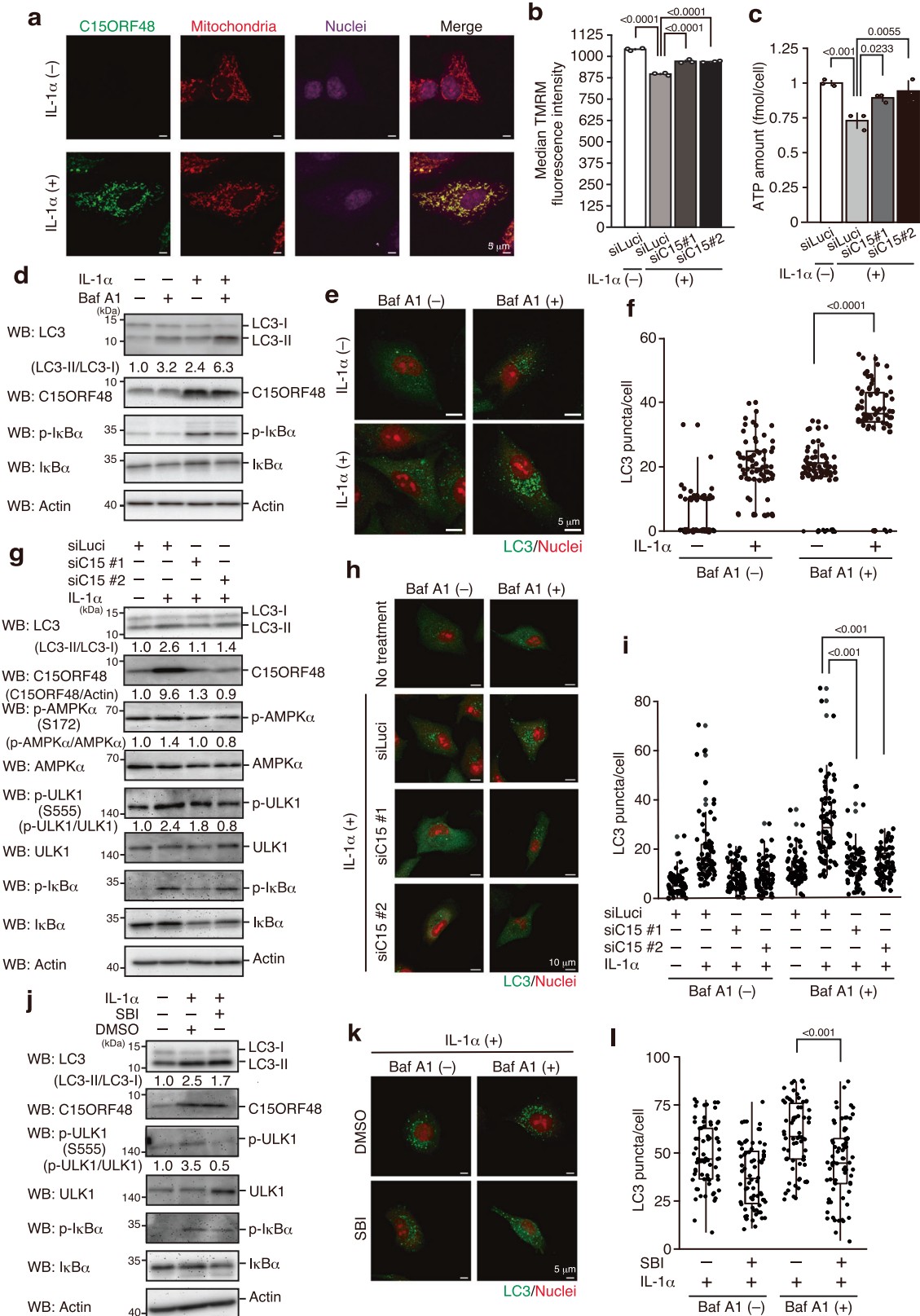

reversed by *ATG5*- or *ATG7*-knockdown (Fig. 5j, k). Thus, these data suggested that IL-1α stimulation induces C15ORF48-dependent autophagy, which promotes glutathione synthesis, and in turn confers resistance to oxidative stress.

We next tested if the same event occurs in MDA-MB-231 tumor cells expressing high levels of C15ORF48. *C15ORF48* knockdown significantly reduced total glutathione and GSH levels (Fig. 5l). Moreover, *C15ORF48* knockdown depressed resistance to oxidative stress and recovered susceptibility to the GPX inhibitor, RSL3 (Fig. 5m−o). Taken together, these results suggest that C15ORF48 induces stress-independent autophagy and promotes cell survival by eliminating oxidative stress via upregulation of glutathione levels.

**Fig. 3 | Induction of endogenous C15ORF48 increases autophagy. a** A549 cells were stimulated with IL-1α (10 ng/mL) or unstimulated for 24 h. After incubation, cells were fixed and stained with anti-C15ORF48 antibody and MitoTracker. Nuclei were counter stained with TO-PRO-3. Representative images are shown from three independent experiments. **b, c** A549 cells were transfected with the indicated siRNAs for 48 h and then stimulated with IL-1α (10 ng/mL) or unstimulated for 24 h. After incubation, median TMRM (**b**) or ATP amount in a cell (**c**) was analyzed and is shown as mean ± SD. Statistical significance was calculated using two-way ANOVA followed by Tukey's multiple comparisons test (*n* = 3, biological replicates). **d, e, f** A549 cells were stimulated or unstimulated with IL-1α (10 ng/mL) for 20 h. After incubation, media were replaced with fresh media with or without IL-1α for an additional 4 h. Half the samples were treated with bafilomycin A1 (Baf A1, 200 μM)

for the final 1 h. After incubation, cells were subjected to western blotting as in Fig. 1d (**d**). Alternatively, cells were fixed and analyzed for LC3 puncta by immunocytochemistry, as in Fig. 2b, c. Statistical significance was calculated using two-way ANOVA followed by Tukey's multiple comparisons test (*n* = 70 cells from two independent experiments) (**e, f**). **g, h, i** As in Fig. 3d, e, f, except that cells were transfected with the indicated siRNAs for 48 before IL-1α stimulation. **j, k, l** A549 cells were stimulated or unstimulated with IL-1α (10 ng/mL) for 20 h. After incubation, media were replaced with fresh media together with or without IL-1α for an additional 4 h. Some samples were treated with DMSO (0.1%) or SBI-0206965 (SBI, 10 μM) for the final 4 h. Cells were used for western blotting (**j**) and LC3 puncta analysis by immunocytochemistry (**k, l**), as in Fig. 3d, e, f.

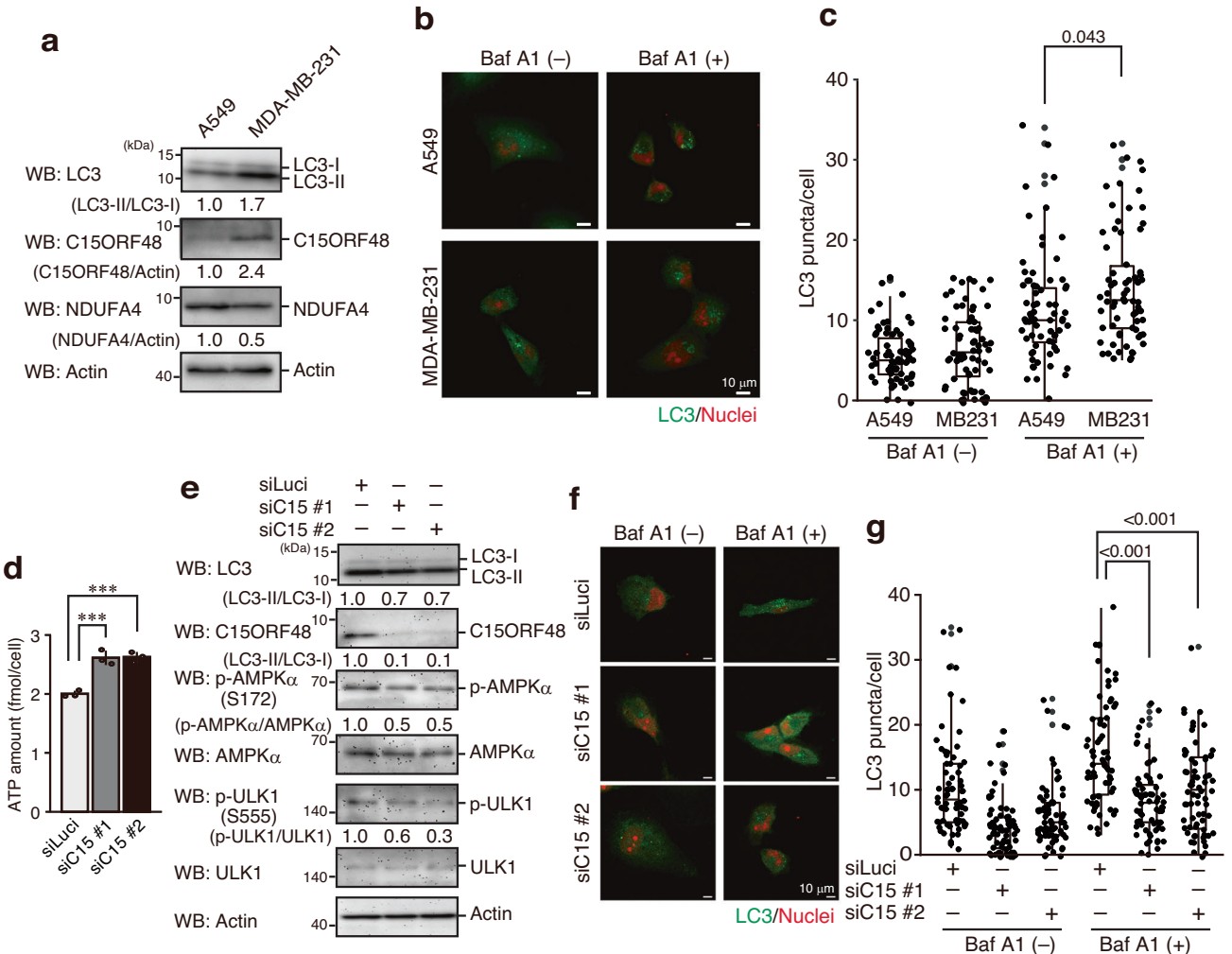

**Fig. 4 | High expression of endogenous C15ORF48 enhances autophagy. a** A549 and MDA-MB-231 cells were lysed and subjected to western blotting with the indicated antibodies. Data are representative of three independent experiments with three biological replicates. **b, c** As in Fig. 2b, c, except that A549 and MDA-MB-231 cells were used. Statistical significance was calculated using two-way ANOVA followed by Tukey's multiple comparisons test (*n* = 70 cells from two independent experiments). **d** MDA-MB-231 cells were transfected with the indicated siRNAs for 48 h. After incubation, cells were used for ATP assays. ATP amount in a cell is shown as means ± SDs. Statistical significance was calculated using two-way ANOVA followed by Tukey's multiple comparisons test (*n* = 3, biological replicates). **e** MDA-

MB-231 cells were transfected with the indicated siRNAs for 48 h. 44 h after transfection, media were replaced with fresh media. After incubation, cells were lysed and subjected to western blotting, as in Fig. 1d. **f, g** MDA-MB-231 cells were transfected with the indicated siRNAs for 48 h. 44 h after transfection, media were replaced with fresh media. Half the samples were treated with bafilomycin A1 (Baf A1, 200 μM) for the final 1 h. Cells were fixed and analyzed for LC3 puncta by immunocytochemistry, as in Fig. 2b, c. Statistical significance was calculated using two-way ANOVA followed by Tukey's multiple comparisons test (*n* = 70 cells from two independent experiments).

As C15ORF48 has two homologs, NDUFA4 and NDUFA4L2 (also known as Coxfa4 and Coxfa4l2, respectively)[22], we assessed whether these homologs have similar effects to C15ORF48 on mitochondrial activity, autophagy, and intracellular metabolism by establishing cell lines stably expressing these proteins. Interestingly, forced expression

of NDUFA4 or NDUFA4L2 did not affect mitochondrial membrane potential, ATP concentrations, basal autophagy activities, or glutathione levels (Supplementary Fig. 3a–g), suggesting that C15ORF48 has an unique role in controlling of mitochondrial activity, autophagy, and intracellular metabolism among the three homologous proteins.

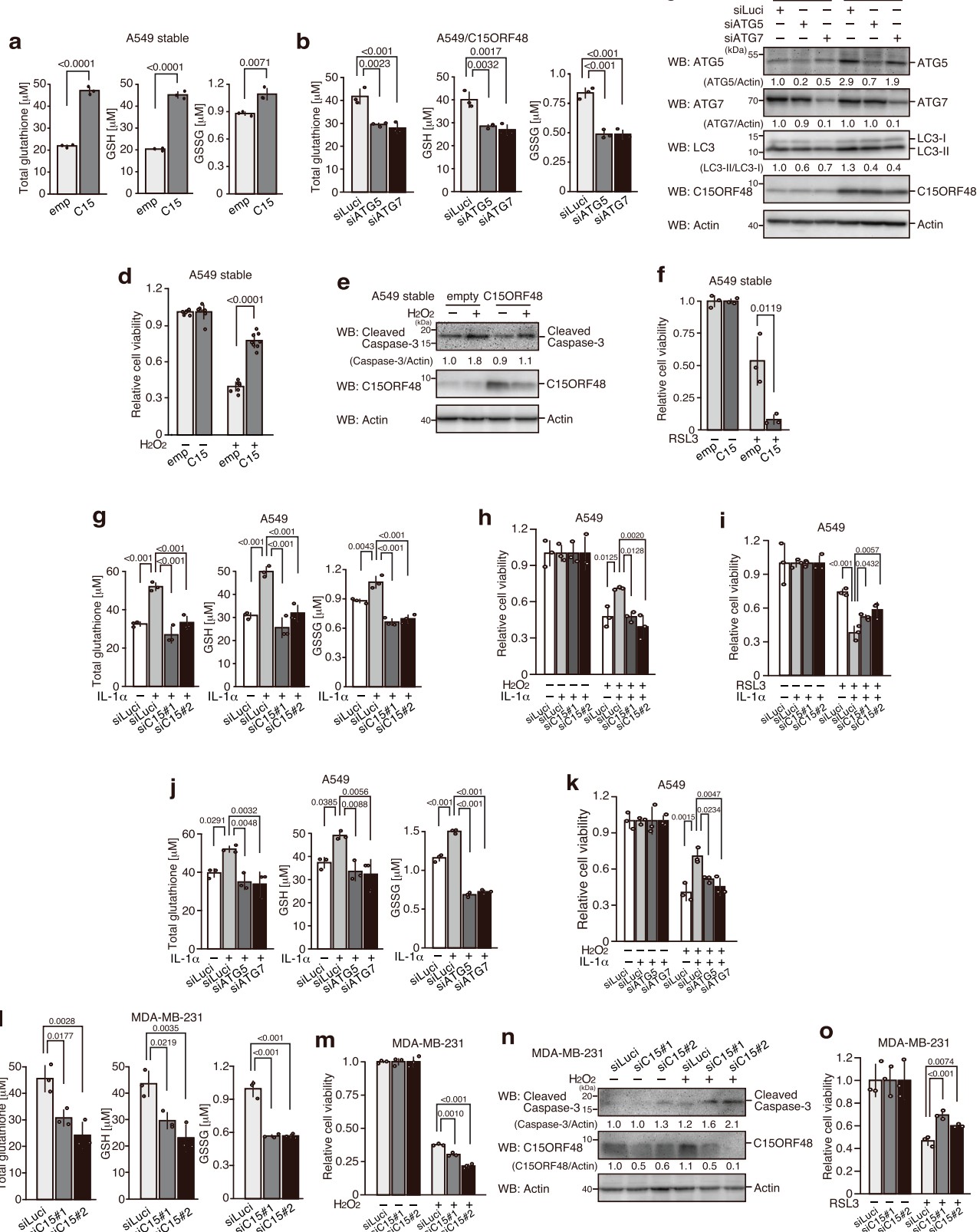

### C15ORF48 is involved in development of TECs and mature CD4 single-positive T cells in the thymus

We next sought to determine the physiological function of C15ORF48-inducing autophagy. Because autophagy is active in TECs in the absence of starvation[12,18], we speculated that C15ORF48 may be involved in constitutive autophagy induction in TECs. We first

investigated expression levels of *C15orf48* mRNA in mouse TECs. Analysis of immunological genome project platform data showed that expression of mouse *C15orf48* (*AA467197*) is high in mature mTECs expressing high levels of MHC class II and co-stimulatory CD80 (mTEC^hi), but low and rare in thymocytes (Supplementary Fig. 4a). Recent single-cell RNA-seq analyses (scRNA-seq) revealed high

**Fig. 5 | C15ORF48-induced autophagy protects cells from oxidative stress.** Glutathione concentrations in A549/empty and A549/C15ORF48 cells (**a**) or A549/C15ORF48 cells transfected with the indicated siRNAs (48 h) (**b**) are shown as means ± SDs. Statistical significance was calculated using two-tailed unpaired Student's *t* test (**a**) or two-way ANOVA followed by Tukey's multiple comparisons test (**b**) ($n = 3$, biological replicates). **c** A549/empty and A549/C15ORF48 cells were transfected with the indicated siRNAs (48 h) and subjected to western blotting, as in Fig. 1d. **d** A549/empty and A549/C15ORF48 cells (20,000 cells/well) were treated with or without $H_2O_2$ (300 μM, 18 h) and subjected to an MTT assay. The quantitative ratio of MTT absorbance in $H_2O_2$-treated cells normalized to that in untreated cells is shown as the mean ± SD. Statistical significance was calculated using two-tailed unpaired Student's *t* test ($n = 8$, biological replicates). **e** A549/empty and A549/C15ORF48 cells were treated or untreated with $H_2O_2$ (300 μM, 6 h) and

subjected to western blotting, as in Fig. 1d. **f** As in Fig. 5d, except that RSL3 (20 μM, 24 h) was used. **g, j** A549 cells were transfected with the indicated siRNAs for 24 h. After incubation, cells were treated with IL-1α (10 ng/mL) or untreated for 24 h and subjected to glutathione assays, as in Fig. 5b. **h, k** As in Fig. 5d, except that A549 cells transfected with the indicated siRNAs (24 h) and treated with or without IL-1α (10 ng/mL, 24 h) were used. Statistical significance was calculated using two-way ANOVA followed by Tukey's multiple comparisons test ($n = 3$, biological replicates). **i** As in Fig. 5h, except that RSL3 (20 μM, 24 h) was used. **l** As in Fig. 5b, except that MDA-MB-231 cells were used. **m** As in Fig. 5h, except that MDA-MB-231 cells were used ($H_2O_2$, 100 μM, 6 h). **n** As in Fig. 5e, except that MDA-MB-231 cells transfected with the indicated siRNAs (48 h) were used. **o** As in Fig. 5m, except that RSL3 (20 μM, 24 h) was used.

heterogeneity of TECs. Thus, in addition to separation of cTECs and mTECs, mTECs are further separated based on expression of AIRE and other marker molecules[40–43]. Re-analysis of scRNA-seq data of mouse TECs[42] suggested expression of mouse *C15orf48* in cTECs (Fig. 6a, cluster 8 and 12), AIRE-expressing (AIRE[+]) mTECs (Fig. 6a, cluster 0 and 2), and a population of post-AIRE mTECs (Fig. 6a, cluster 9, 10, and 14), which are defined as AIRE-negative mTECs, differentiated from AIRE[+] mTECs. In contrast, *C15orf48* expression is minimal in immature mTECs (Fig. 6a, cluster 3, 4, and 5) and transit-amplifying mTECs (Fig. 6a, cluster 1), which are precursors of AIRE[+] mTECs[42,44]. We further performed flow cytometric analysis to confirm C15ORF48 protein expression by using anti-mouse C15ORF48 antibody. Because of relatively high non-specific binding of this antibody in flow cytometric analysis, the mean fluorescent intensity (MFI) for antibody binding in wild-type TECs was subtracted with that in TECs from *C15orf48*-deficient (*C15orf48*[−/−]) mice (Fig. 6b–e and Supplementary Fig. 4b, c). Importantly, whereas C15ORF48 protein is abolished by the CRISPR-Cas9 system in these mutant mice, the expression level of NDUFA4 protein, which is regulated by miR-147b encoded in the 3′-UTR of the *C15orf48* gene[23,24,26,27], was not affected (Supplementary Fig. 4d). Furthermore, we analyzed the effects of C15ORF48-deficiency on mitochondrial activities using primary embryonic fibroblasts from *C15orf48*[−/−] mice. *C15orf48*[−/−] fibroblasts showed upregulation in mitochondrial potential and ATP levels and in turn downregulation in AMPKα-ULK1 activation and basal autophagy activity. These mutant fibroblasts also showed reduction in glutathione levels and resistance to oxidative stress (Supplementary Fig. 5a–f), indicating that C15ORF48 has a suppressive role in mitochondrial activities and thereby enhances autophagy and resistance to oxidative stress in untransformed primary cells.

TECs (EpCAM[+]CD45[−]TER119[−] cells) were separated into cTECs and mTECs, based on expression of Ly51 and binding of UEA-1 lectin (Fig. 6b). mTECs (UEA-1[+]Ly51[−]) were further divided into mTECs[lo] and mTECs[hi] by the expression level of CD80 (Fig. 6b). C15ORF48 protein expression was detected in mTECs[hi] and cTECs, but not mTECs[lo] (Fig. 6b, c). qPCR analyses showed high expression of *C15orf48* mRNA in mTECs[hi] and cTECs, but not in mTECs[lo] (Supplementary Fig. 6). mTECs[hi] were further sub-classified into Early-, Late-, and Post-Aire mTECs based upon CD24 and Sca1 expression[45]. Flow cytometric analysis showed C15ORF48 protein expression in all of these subpopulations (Fig. 6d, e). Consequently, C15ORF48 is expressed in mature types of TECs, implying its functions in self-antigen presentation through autophagy-dependent self-protein degradation.

We next investigated influences of C15ORF48 depletion in thymic cell development. The weight of the thymus and total thymic cell number did not differ significantly between wild-type and *C15orf48*[−/−] mice (Supplementary Fig. 7a). Numbers of cTECs and Post-Aire mTECs were slightly reduced in *C15orf48*[−/−] mice (Fig. 6f). This suggests that C15ORF48 participates in maintenance of cTECs and Post-Aire mTECs, which suggests a function of C15ORF48 in cell survival.

In the thymus, CD4 and CD8 single-positive thymocytes (CD4SP and CD8SP) differentiate from bone marrow-derived progenitors via the stage of CD4 and CD8 double-positive thymocytes (DP)[46,47]. Flow cytometric analysis using CD4 and CD8 antibodies suggested that numbers and ratios of each subset to total thymocytes were not significantly affected in *C15orf48*[−/−] mice (Supplementary Fig. 7b). Positive selection of thymocytes was further investigated on the basis of CD3 and CD69 expression levels[48]. Numbers and percentages of immature CD3[lo] CD69[lo] cells (population 1; mostly CD4 and CD8 double-negative (DN) and pre-selection DP thymocytes), CD3[int] CD69[lo] cells (population 2; mostly pre-selection DP cells), CD3[int] CD69[hi] cells (population 3), CD3[hi] CD69[hi] cells (population 4; post-positive selection thymocytes), and CD3[hi] CD69[lo] cells (population 5; mature SP cells) were not changed in *C15orf48*[−/−] mice (Supplementary Fig. 8a). Negative selection of thymocytes and Tregs were examined by detecting expression of Helios[49] and expression of CD25 and Foxp3[50], respectively. Numbers and percentages of Helios[+] cells (post-negative selection thymocytes) and CD25[+] Foxp3[+] cells (Tregs) were largely unaffected in *C15orf48*[−/−] mice (Supplementary Fig. 8b, c). CD4SPs and CD8SPs in the thymic medulla are further subdivided into three subpopulations, according to their maturation levels: semi-mature (SM), mature 1 (M1), and mature 2 (M2)[51]. Intriguingly, the ratio of the M2 subfraction in CD4SP cells was slightly reduced in *C15orf48*[−/−] mice (Fig. 6g) whereas no apparent change was detected in the number or ratio of CD8SP cells (Supplementary Fig. 9). Thus, C15ORF48 may contribute to the development of mature CD4SP cells, yet it may not be indispensable for the overall process of thymic T cell selection.

## C15ORF48 is critical for stress-independent autophagy induction in TECs

We addressed the requirement of C15ORF48 for stress-independent autophagy in TECs. *C15orf48*[−/−] mice were crossed with mice expressing green fluorescent protein (GFP)-labeled LC3 (GFP-LC3) to monitor autophagosome formation[12]. As reported previously, GFP-LC3 puncta were detected in the thymus of control mice without starvation (Fig. 7a, b). Co-immunostaining showed that GFP puncta are formed in cells expressing Keratin-5 (mTECs) or Keratin-8 (cTECs), confirming high autophagy activity in mTECs and cTECs without starvation (Fig. 7a, b). Strikingly, the number of GFP-LC3 puncta in mTECs is significantly diminished in the absence of C15ORF48 (Fig. 7a). In cTECs, the number of GFP-LC3 puncta is slightly but significantly reduced in *C15orf48*-deficient thymus (Fig. 7b). In contrast, GFP-LC3 puncta emerging during fasting are comparable in skeletal muscles between controls and *C15orf48*[−/−] mice, suggesting that starvation-induced autophagy is not affected by depletion of C15ORF48 (Fig. 7c). Taken together, these results indicate that C15ORF48 controls stress-independent autophagy in mTECs and, to a lesser extent, cTECs, but not typical starvation-induced autophagy.

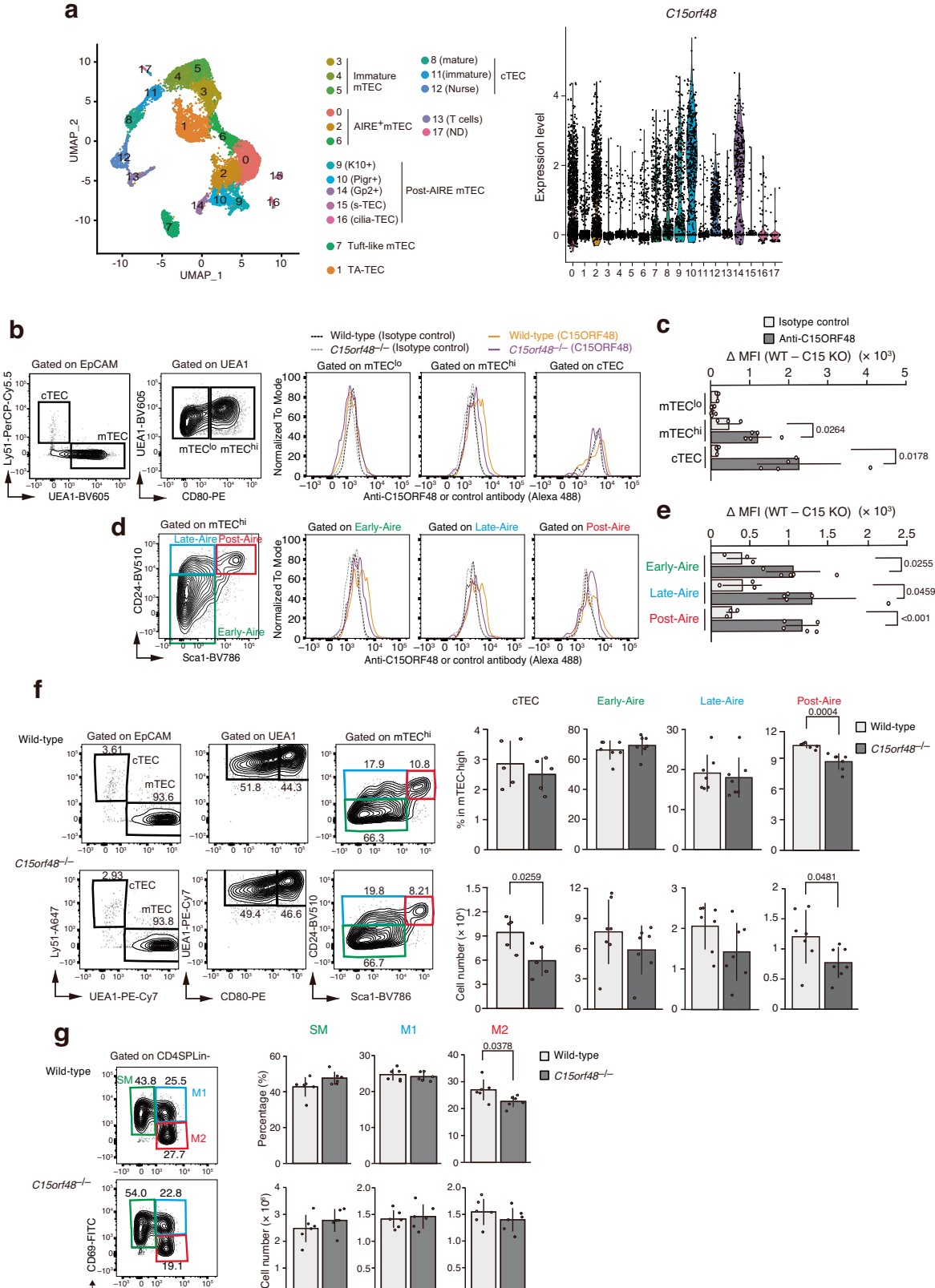

## C15ORF48 is required for prevention of autoimmunity

Because autophagy in TECs contributes to T cell self-tolerance, we wondered whether *C15orf48*⁻/⁻ mice exhibit signs of autoimmunity. Although no apparent changes were observed in cell numbers or ratios of effector memory T cells or Tregs in lymph node T cells of *C15orf48*⁻/⁻ mice (Supplementary Fig. 10a–e), sera from *C15orf48*⁻/⁻ mice showed high immune reactivities against mouse tissue sections, such as lacrimal glands, submandibular glands, eyes, and ovaries (Fig. 8a). Sera from these mutant mice also showed high immune reactivities against mouse tissue lysates from lung, kidney, and liver in immunoblotting (Fig. 8b, Supplementary Fig. 11a), suggesting that sera from *C15orf48*⁻/⁻ mice contain autoantibodies reactive to self-antigens in these organs.

**Fig. 6 | C15ORF48 is expressed in cTECs and mature mTECs and is involved in development of TECs and mature CD4 single-positive T cells in the thymus. a** Uniform manifold approximation and production (UMAP) plot of scRNA-seq data from TECs (EpCAM⁺ CD45⁻ TER119⁻) from 4-week-old mice. Cell clusters (R0 to R17) are indicated by colors and numbers in the plot (left). Violin plots depicting expression levels of *C15orf48* in each cluster (right). **b, c** Representative images of flow cytometry plots of cTECs, mTEC^lo and mTEC^hi in wild-type mice (**b**, left). Representative images of MFIs of anti-C15orf48 or isotype control antibodies in mTEC^lo, mTEC^hi, and cTEC from 4-week-old wild-type and *C15orf48*⁻/⁻ mice (**b**, right). MFIs of anti-C15orf48 or isotype control antibodies in TECs from wild-type mice were subtracted with those in TECs from *C15orf48*⁻/⁻ mice and are shown as means ± SDs. Statistical significance was calculated using two-way ANOVA followed by Tukey's multiple comparison test (*n* = 5, biological replicates) (**c**). **d, e** As in

Fig. 6b, c, except that data of Early-Aire, Late-Aire, and Post-Aire mTEC subfractions are shown, instead of those of cTECs, mTEC^lo and mTEC^hi. **f** Representative images of flow cytometry plots of cTECs, mTEC^lo, mTEC^hi and the three mTEC subfractions in 4-week-old wild-type and *C15orf48*⁻/⁻ mice (left). Numbers of cTECs, Early-Aire, Late-Aire, and Post-Aire mTECs and their ratios to total thymic cells are shown as means ± SDs. Statistical significance was calculated using two-tailed unpaired Student's *t* test (*n* = 5 (cTECs); *n* = 7 (mTECs), biological replicates) (right). **g** Representative images of flow cytometry plots of SM, M1, and M2 CD4SP cells in 4-week-old wild-type and *C15orf48*⁻/⁻ mice (left). Numbers of SM, M1, and M2 CD4SP cells and their ratios to total thymocytes are shown as means ± SDs. Statistical significance was calculated using two-tailed unpaired Student's *t* test (*n* = 5, biological replicates) (right).

Remarkably, enhanced infiltration of inflammatory cells was observed in lung, kidney, and liver sections from *C15orf48*⁻/⁻ mice (Fig. 8c). Furthermore, kidney sections from *C15orf48*⁻/⁻ mice exhibited increased deposits of IgG in glomerulus-like structures (Fig. 8d, Supplementary Fig. 11b).

To verify whether autoimmune phenotypes in *C15orf48*⁻/⁻ mice depend on C15ORF48-deficiency in thymic stroma cells including TECs, we transplanted *C15orf48*⁻/⁻ fetal thymic stroma, which was obtained by depleting hematopoietic cells from the embryonic thymus, into the renal capsules of athymic nude mice[15]. Eight weeks later, successful engraftment was observed in the nude mice grafted with *C15orf48*⁻/⁻ thymus (KO/nu) as well as in the nude mice grafted with control littermate thymus (WT/nu) (Fig. 9a). Sera from KO/nu mice showed high immune reactivities against mouse tissue sections of lacrimal glands, submandibular glands, eyes, ovaries, lungs, kidneys, and livers (Fig. 9b). Furthermore, enhanced infiltration of inflammatory cells was observed in tissue sections from lacrimal glands, submandibular glands, lungs, kidneys, and livers from KO/nu mice (Fig. 9c), suggesting that the engraftment of *C15orf48*-deficient thymus in nude mice causes autoimmunity. Overall, these results indicate that C15ORF48 in TECs contributes to establishing self-tolerance (Fig. 10).

## Discussion

In this study, we found that the mitochondrial protein, C15ORF48, activates autophagy (Fig. 10). Autophagy is generally accepted as a phenomenon induced by cellular stress, such as starvation or mitochondrial damage, and prevents cell death by promoting recycling of nutrients or eliminating damaged organelles. In contrast to stress-dependent autophagy, C15ORF48-induced autophagy is independent of cellular stress for the following three reasons: (1) autophagy is activated even in non-starved conditions, confirming that mTOR phosphorylation was not reduced by C15ORF48 expression, (2) mitophagy is not activated in C15ORF48-expressing cells, and (3) C15ORF48 is required for constitutive autophagy in TECs and is dispensable in starvation-dependent autophagy in mouse skeletal muscle.

Our results show that C15ORF48 reduces mitochondrial membrane potential and lowers intracellular ATP levels, thereby leading to autophagy via AMPK-ULK1 signaling. Repression of mitochondrial membrane potential causes mitochondrial stress and subsequent mitophagy[52,53], whereas mitophagy did not increase in C15ORF48-expressing cells. Therefore, C15ORF48 is likely to alleviate, rather than exacerbate, mitochondrial activation and to promote cell survival by inducing stress-independent autophagy.

Furthermore, our present data show that C15ORF48-induced autophagy increases intracellular glutathione levels and prevents oxidative stress. Glutathione is a tripeptide composed of three amino acids (glutamic acid, cysteine, and glycine) and acts as an important cofactor for the antioxidant enzyme, GPX[54]. C15ORF48-expressing cells showed higher susceptibility to a GPX inhibitor, suggesting that C15ORF48-expressing cells strengthen the glutathione-GPX antioxidant system for survival. Given that nonselective macroautophagy

is involved in recycling of nutrients[1,3], C15ORF48-induced autophagy may be classified as nonselective macroautophagy that facilitates degradation of proteins and recycling of amino acids for glutathione synthesis.

In the thymus, *C15orf48* is expressed in mTECs and cTECs, but its regulation in these cells remains unclear. It is known that NF-κB signaling is pivotal in TEC differentiation. Differentiation of mTECs is controlled by the TNF receptor superfamily member receptor activator of NF-κB (RANK) and CD40[55–57], the NF-κB component RelB[58], NF-κB-inducing kinase (NIK)[59], and NF-κB signal transducer TNF receptor-associated factor 6 (TRAF6)[60]. Mice harboring mutations of these genes show defects in mTEC differentiation. NIK is also involved in cTEC differentiation[61]. Given that NF-κB is a key transcription factor for *C15ORF48* expression in human cancer cell lines, NF-κB signaling is likely to be involved in *C15orf48* expression in mTECs and cTECs.

In the present study, *C15orf48*⁻/⁻ mice showed reduced autophagic activity in mTECs and, to a lesser extent, cTECs. Furthermore, *C15orf48*⁻/⁻ mice showed a reduction in the M2 subfraction of CD4SP cells and autoimmune phenotypes such as the increment of auto-antibodies in sera, deposits of IgG in kidneys, and infiltration of inflammatory cells into tissues. Autophagy in mTECs is important for negative selection of CD4SP cells because autophagy substrates are loaded on MHC class II and preferentially presented to CD4SP thymocytes as self-antigens[18,19,62]. Autophagy in cTECs is also involved in presentation of self-antigens to thymocytes and following positive selection and development of CD4SP cells[62,63]. Despite the fact that *C15orf48*⁻/⁻ mice showed reduction in autophagy in mTECs and cTECs, no apparent difference was observed in cell numbers and percentages of thymocytes during positive or negative selection and Tregs. Our interpretation of these findings suggests that the autophagy-dependent processing of self-antigens by C15ORF48 may play a role in shaping only a subset of the TCR repertoires, rather than governing the entire TCR selection process. To explore this notion further, a comprehensive analysis of TCR repertoires in mature thymic T cells and studies using TCR transgenic mice should be a subject for future research. Notably, a difference in the C15ORF48 dependency between cTECs and mTECs in autophagy induction was suggested, which may also explain the limited impact observed in thymic positive selection by the C15ORF48 deficiency. In addition, *C15orf48*⁻/⁻ mice showed a slight reduction in mTECs (Post-Aire) and cTECs, suggesting that C15ORF48-induced autophagy is involved in maintenance of these cells via suppression of basal oxidative stress. Therefore, it is possible that a reduction of TECs affects differentiation and selection of CD4SP cells in *C15orf48*⁻/⁻ mice.

Recent studies have shown that $H_2O_2$-mediated redox status is closely associated with basal autophagy and negative selection of thymocytes in TECs using transgenic mice overexpressing the hydrogen peroxide quenching enzyme, catalase[21]. Although the mechanism by which $H_2O_2$ regulates autophagy in TECs is unclear, AMPK-ULK1 signaling is a candidate target of $H_2O_2$ in controlling autophagy[64]. Because AMPK-ULK1 signaling is activated downstream of C15ORF48,

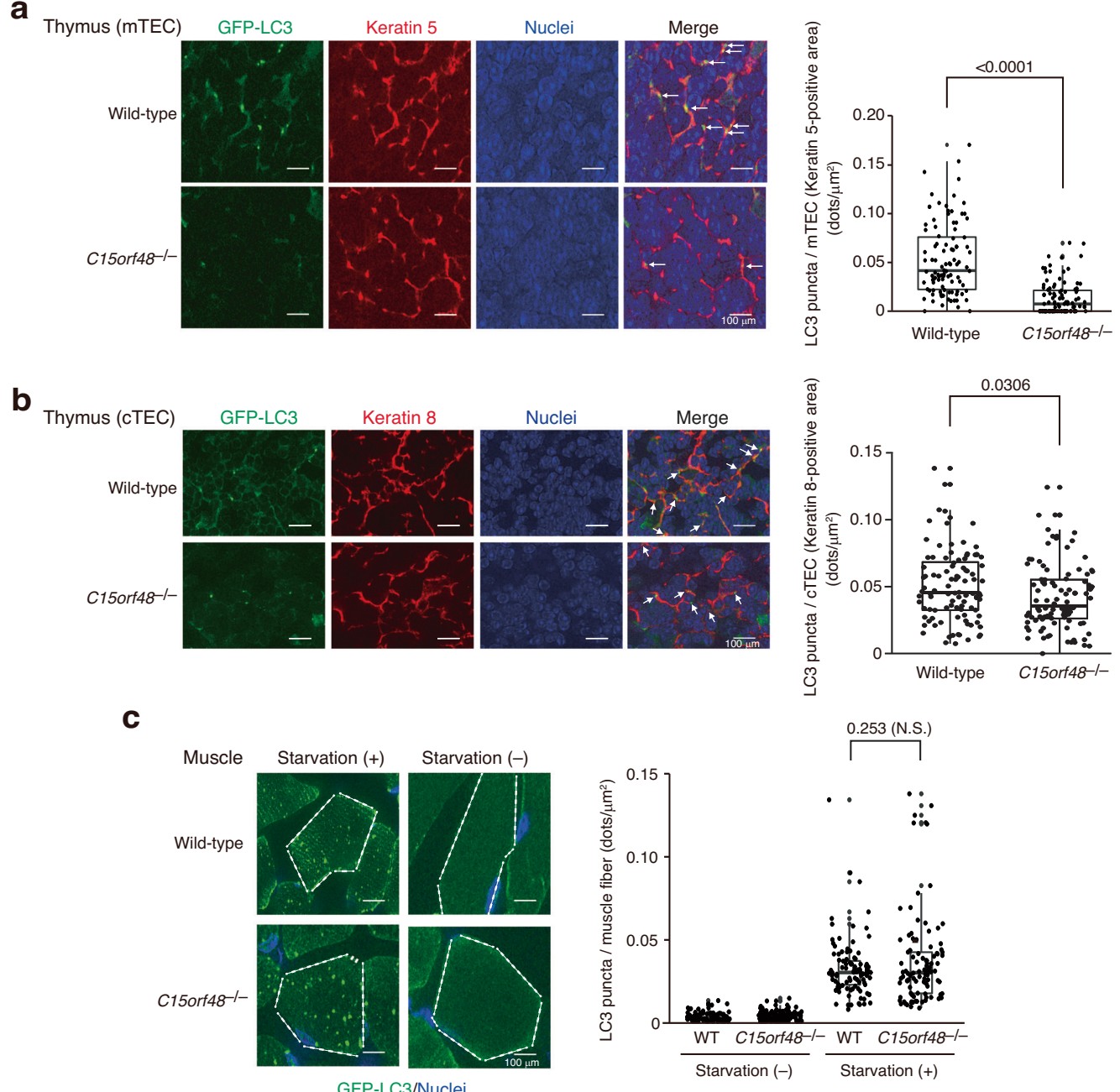

**Fig. 7 | C15ORF48 is critical for stress-independent autophagy in TECs.** Thymus sections from 4-week-old normal fed wild-type and *C15orf48*[−/−] mice were stained with anti-GFP and anti-Keratin 5 (**a**) or anti-Keratin 8 (**b**) antibodies and DAPI (left). GFP-LC3 puncta in Keratin 5- or 8-positive areas are indicated by arrows (left panel). Numbers of GFP-LC3 puncta in Keratin 5-positive areas are shown (right panel). Statistical significance was calculated using two-tailed unpaired Student's *t* test (*n* = 100 sections derived from three independent mice). **c** GFP-LC3 mice (wild-type) and GFP-LC3/*C15orf48*[−/−] (*C15orf48*[−/−]) mice (4-week-old) were starved for 48 h. After fasting, thigh skeletal muscle sections were prepared and stained with an anti-GFP antibody and DAPI. Single muscle fibers are encircled by broken lines (left panel). The number of GFP-LC3 puncta in the muscle area is shown (right panel). Statistical significance was calculated using two-tailed unpaired Student's *t* test (*n* = 100 sections derived from three independent mice). Wild-type, WT. Not significant, N.S.

this signaling is likely to be a key mediator of constitutive autophagy in TECs.

Cancer cells exhibit constitutive autophagy, which enhances their survival and proliferation[10]. However, molecular mechanisms underlying starvation-independent autophagy in cancer cells remain largely unknown. In this study, we showed that human breast cancer (MDA-MB-231) cells have high expression of C15ORF48 and high basal autophagy activity. Notably, C15ORF48 increases glutathione levels and reduces oxidative stress in these cells. MDA-MB-231 cells have

constitutive activation of NF-κB[38], which is frequently observed in various types of cancer cells and is important in cancer progression. Because constitutively active NF-κB in cancer cells drives *C15ORF48* expression[65], C15ORF48 may be pivotal in constitutive autophagy and may promote cancer cell survival by preventing basal oxidative stress.

In this study, we found that *C15orf48*[−/−] mice show reduced stress-independent autophagy in mTECs and exhibit autoimmunity, suggesting that C15ORF48 is involved in negative selection of thymocytes and acquisition of self-tolerance. *C15orf48*[−/−] mice have autoantibodies

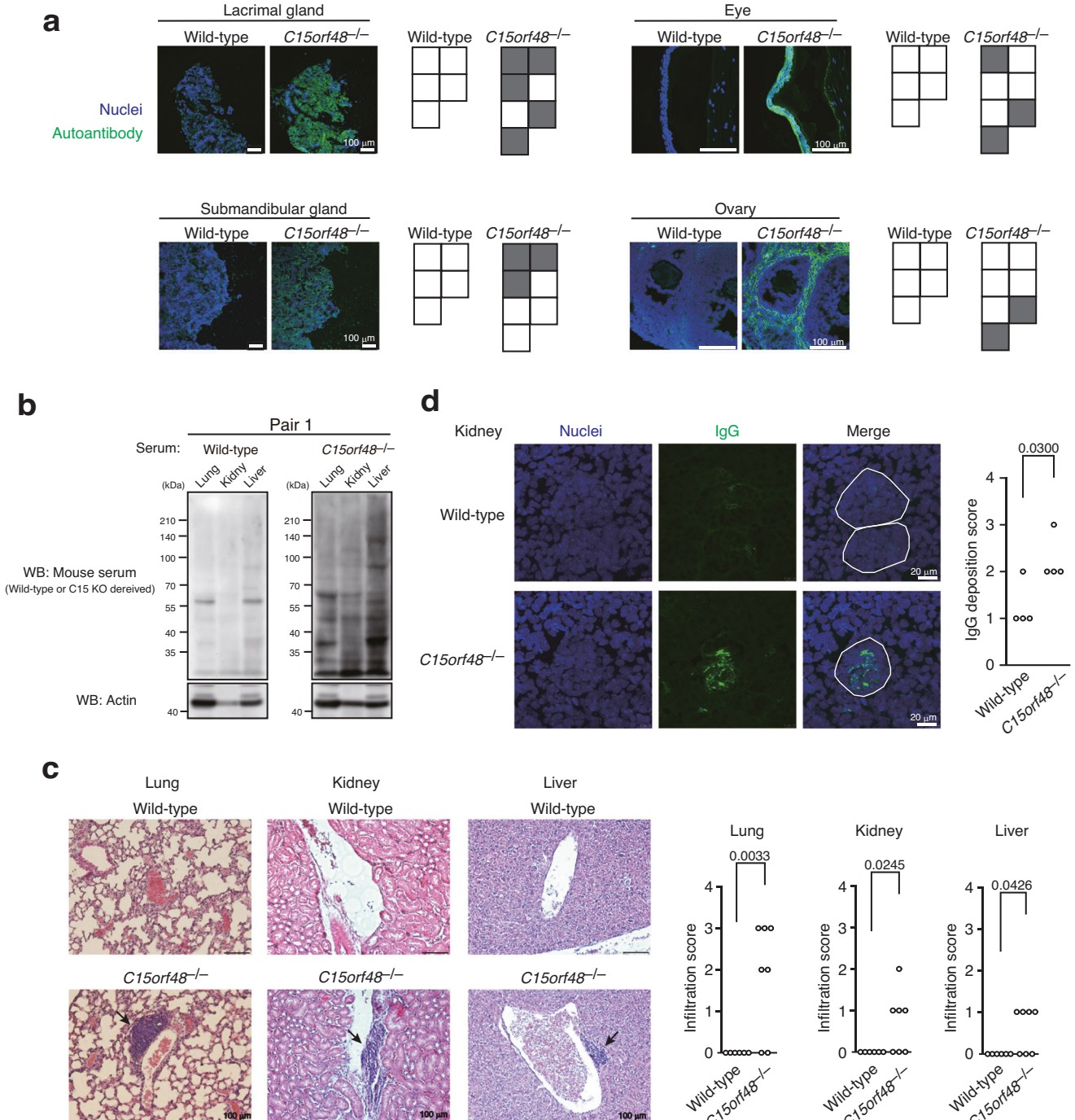

**Fig. 8 | C15ORF48 is required for prevention of autoimmunity in mice.**
**a** Immunostaining of tissue sections from *Rag1*[−/−] mice (C57BL/6 background, 8–10-week-old) with sera from 21-week-old wild-type and *C15orf48*[−/−] mice (C57BL/6 background). Nuclei were counter-stained with DAPI. Each box represents serum from a single mouse (wild-type, *n* = 5 mice; *C15orf48*[−/−], *n* = 7 mice). The gray box indicates the detection of reactivity in tissue sections from *Rag1*[−/−] mice. **b** Western blotting of tissue lysates from *Rag1*[−/−] mice with sera from 21-week-old wild-type and *C15orf48*[−/−] littermate mice. All sera were diluted equally at 1:1000 with 5% skim-milk containing tris buffer. Western blotting using these sera was performed simultaneously to enable exact comparison of their immunoreactivities. Anti-actin blots were used as loading controls. Data are representative of five independent experiments with sera from five different pairs of littermate mice. **c** Hematoxylin and eosin staining of lung, kidney, and liver sections from 21-week-old wild-type

and *C15orf48*[−/−] mice. Inflammatory cell infiltrations are indicated by arrows (left). Infiltration scores were determined based on hematoxylin and eosin staining scored from 0 to 4, in a double-blind manner. Each point represents a value from an individual mouse. Statistical significance was calculated using two-tailed unpaired Student's *t* test (wild-type, *n* = 6 mice; *C15orf48*[−/−], *n* = 7 mice) (right). **d** Immunostaining of kidney sections from 21-week-old wild-type and *C15orf48*[−/−] mice with anti-mouse IgG. Nuclei were counter-stained with DAPI. Glomerulus-like cell clusters are encircled by white lines. (left). IgG deposition scores were determined based on IgG fluorescence intensities scored from 0 to 4, in a double-blind manner. Each point represents a value from an individual mouse. Statistical significance was calculated using two-tailed unpaired Student's *t* test (wild-type and *C15orf48*[−/−], *n* = 4 mice) (right).

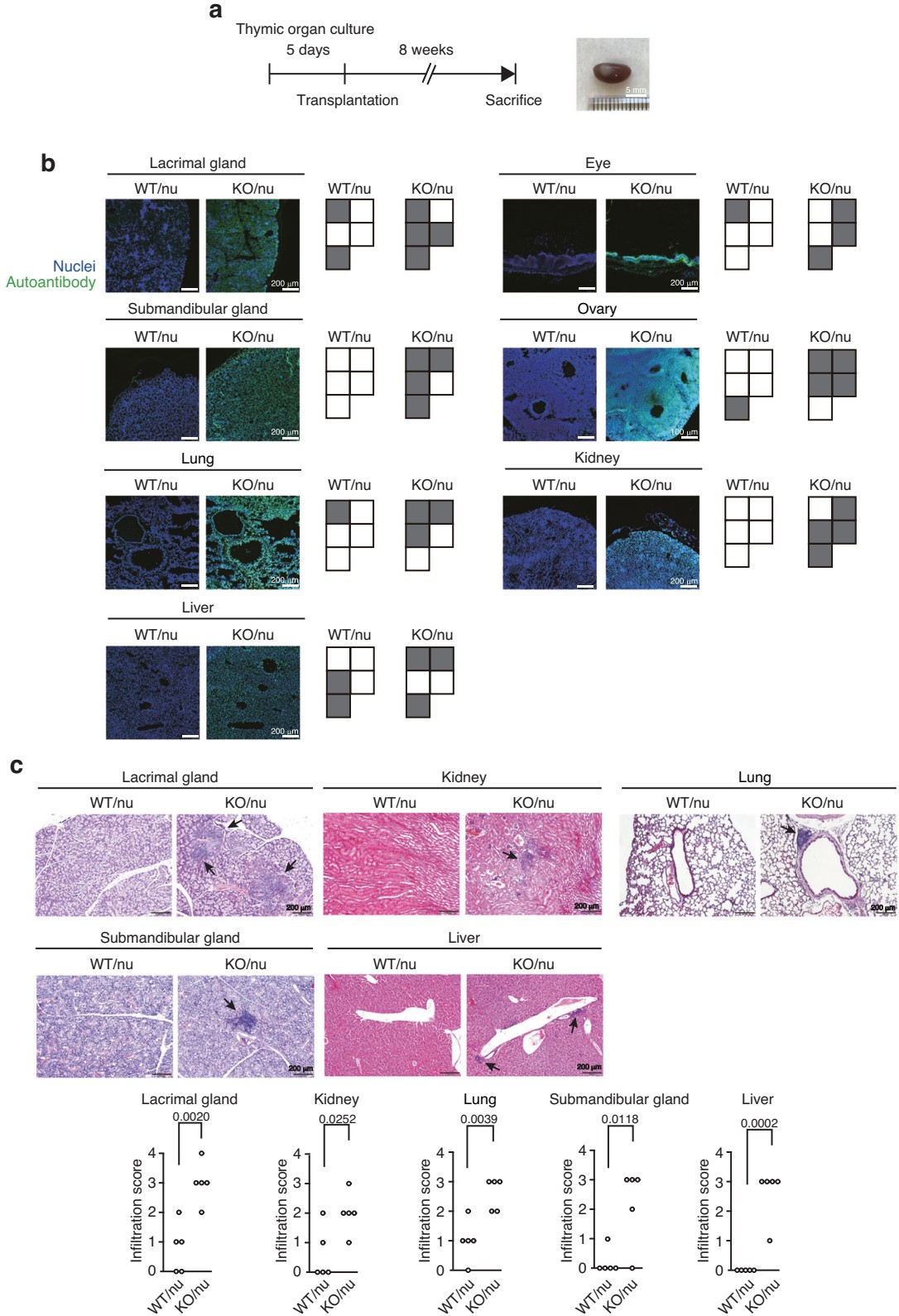

**Fig. 9 | C15ORF48 deficiency in thymic stroma causes autoimmunity. a** A schematic representation of thymic stroma transplantation experiments (left). A representative image of thymus re-generated into a renal capsule of a nude mouse eight weeks after the transplantation (right). **b** Immunostaining of tissue sections from *Rag1*[−/−] mice with sera from WT/nu and KO/nu mice eight weeks after the transplantation. Nuclei were counter-stained with DAPI. Each box represents serum from a single mouse (WT/nu and KO/nu, *n* = 5 mice). The gray box indicates the detection of reactivity in tissue sections from *Rag1*[−/−] mice. **c** Hematoxylin and eosin staining of tissue sections from WT/nu and KO/nu mice eight weeks after the transplantation. Inflammatory cell infiltrations are indicated by arrows (top). Infiltration scores were determined based on hematoxylin and eosin staining scored from 0 to 4, in a double-blind manner. Each point represents a value from an individual mouse. Statistical significance was calculated using two-tailed unpaired Student's *t* test (WT/nu and KO/nu, *n* = 5 mice) (bottom).

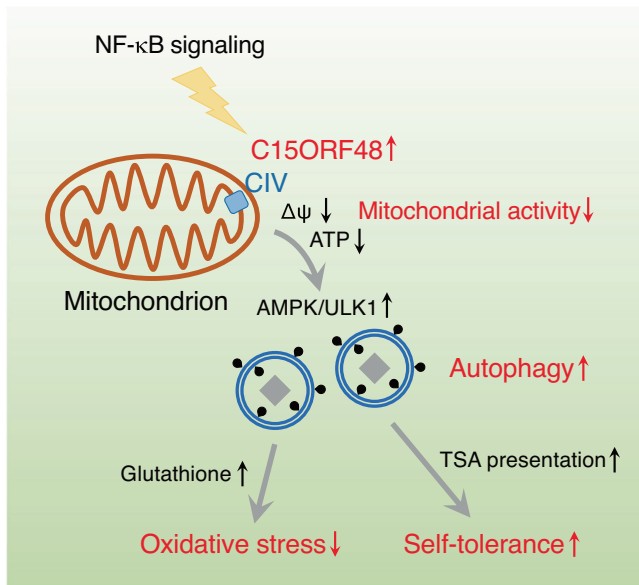

**Fig. 10 | A schematic model of C15ORF48-induced, stress-independent autophagy.** The mitochondrial protein, C15ORF48, a subunit of electron transport chain complex IV, is highly expressed in several cancer cells and thymic epithelial cells, and NF-κB signaling is important for its expression. C15ORF48 reduces mitochondrial activity and intracellular ATP levels, thereby inducing autophagy via proautophagic AMPK-ULK1 signaling independently of starvation or mitochondrial stress. C15ORF48-induced autophagy in cancer cells increases intracellular glutathione levels and prevents oxidative stress. C15ORF48-induced autophagy in thymic epithelial cells regulates self-tolerance and prevents autoimmunity.

against eyes and ovaries, which were also observed in *Aire*-deficient mice[15]. These observations support the notion that C15ORF48 is involved in AIRE-dependent negative selection of thymocytes. *AIRE* was identified as the primary causative gene in human patients suffering from an autoimmune disease[66–68]. Autoantibodies against lacrimal glands, submandibular gland, and eyes, which were detected in *C15orf48*[−/−] mice, are pathological characteristics of autoimmune diseases, such as Sjögren's syndrome[69–71]. Therefore, it is possible that inactivation of the *C15ORF48* gene causes human autoimmune diseases. Mutations and polymorphisms in the *C15ORF48* gene in autoimmune disease patients should be analyzed in the future.

## Methods
### Mouse models
All animal experiments were approved by the Institutional Animal Care and Use Committee of RIKEN Yokohama Branch (2018-075). All mice were maintained under pathogen-free conditions and handled in accordance with Guidelines of the Institutional Animal Care and Use Committee of RIKEN, Yokohama Branch. Almost all available mutant and control mice were randomly used for experiments without selection. Only female mice were used for immunological experiments to exclude the effects of sex difference on immunological phenotypes. Mice were maintained under 12-h light/12-hour dark cycle. The room temperature was regulated at 20 °C and humidity was controlled at 50%. All mice were sacrificed by carbon dioxide inhalation.

C57BL/6 mice, BALB/c mice, and BALB/c nude mice were purchased from CLEA. Littermates and age-matched wild-type mice were used as controls. GFP-LC3 mice (RBRC00806) were obtained from RIKEN BRC. C57BL/6 *Rag1*[−/−] mice were obtained from the Department of Animal Management at RIKEN.

*C15orf48* knockout mice were generated using the CRISPR-Cas9 system with fertilized eggs derived from C57BL/6J mice. C57BL/6J female mice were injected with 5 IU Pregnant Mare Serum

Gonadotropin (PMS). After 48 h, the mice were injected with 5 IU human chorionic gonadotropin (hCG). Serially injected female mice were mated with C57BL/6J male mice. Fertilized eggs were isolated from mating plugs of mated female mice. Synthetic crRNA (Alt-R CRISPR-Cas9 crRNA, IDT, 16 ng/mL), tracrRNA (Alt-R CRISPR-Cas9 tracrRNA, IDT, 24 ng/mL), and Cas9 nuclease (Alt-R S.p. HiFi Cas9 Nuclease V3, 100 ng) were dissolved in 100 mL of Opti-MEM I and electroporated into the embryos. The sequence of the crRNA was as follows: 5′-ACGCUUAUAAAAAAACGCCAAGUUUUAGAGCUAUGCU-3′. Electroporation was performed as described previously[72]. To obtain newborn mice, 2-cell stage embryos were transferred to oviducts of pseudopregnant ICR females.

Genomic DNA of offspring was extracted from tail biopsies using 50 mM NaOH and incubated for 20 min at 95 °C. Extracted samples were chilled on ice and then mixed with 1 M Tris-HCl (pH 8.0). After mixing, samples were centrifuged at 4 °C and collected supernatant was used for genotyping PCR. The following three primers were used for PCR: 5′-CTCAGCTCATTCCTTTGGCG-3′ (sense, F1), 5′-GGTCCG GGGGAAGATGTTTT-3′ (sense, F2), and 5′-GGCTCGAACTGTGTAGGC AT-3′ (antisense, R). After initial denaturation at 94 °C for 2.5 min, PCR was performed for 30 cycles (15 s at 94 °C, 15 s at 55 °C, and 30 s at 72 °C), followed by 5 min additional extension at 72 °C using the BIO-TAQ DNA polymerase (NIPPON Genetics). While 618-bp DNA fragments were amplified from the wild-type genomic DNA with primer pairs F1 and R, 836-bp DNA fragments were amplified from *C15orf48*[−/−] genomic DNA with primer pairs F2 and R.

### Plasmids
The cDNA fragment encoding *C15ORF48* (lacking the 5′ or 3′ UTR) was generated by PCR from the reverse-transcribed product of A549 cells total RNA. The following primers were used for PCR:5′-TAGCTTAAG CCACCATGAGCTTTTTCCAACTCCTGATGAAAAGG-3′ (sense) and 5′-T CAGAATTCTCATTTGGTCACCCTTTGGACATTTTGCAA-3′ (antisense). The synthesized cDNA fragment was inserted into the AflII and EcoRI sites of the pIRESpuro3-CAG vector[73]. Expression plasmids for *NDUFA4* (OHu10657D) and *NDUFA4L* (OHu03212D) were obtained from Genscript.

### Cell culture and antibodies
A549 and MDA-MB-231 cell lines were purchased from ATCC and cultured in Dulbecco's modified Eagle's medium (DMEM) supplemented with 5% fetal bovine serum (FBS) at 37 °C. To establish stable cell pools expressing C15ORF48, A549 cells were transfected with the pIRESpuro3-CAG/C15ORF48 vector or an empty vector using Lipofectamine2000 (Thermo Fisher Scientific) following the manufacturer's instructions. Transfected cells were selected for puromycin resistance. To establish stable cell pools expressing NDUFA4 or NDUFA4L2, A549 cells were transfected with the pcDNA3/NDUFA4 (OHu10657D), pcDNA3/NDU-FA4L2 (OHu03212D) or an empty vector using Lipofectamine2000, and then transfected cells were selected for G418 resistance. siRNAs (5 nM) were transfected using the reverse transfection method with Lipofectamine RNAiMax (Thermo Fisher Scientific), according to the manufacturer's instructions. Silencer select siRNAs specific to *C15ORF48* #1 (s38981), *C15ORF48* #2 (s228367), *ATG5* (s18160), and *ATG7* (s20650) were purchased from Thermo Fisher Scientific. siRNA for luciferase (sense:5′-GCGCUGCUGGUGCCAACCCTT-3′ and antisense:5′-GGGUUG GCACCAGCAGCGCTT-3′) was purchased from Hokkaido System Sciences and used as a control siRNA. Recombinant human IL-1α and TNF was purchased from PeproTech. The IKK inhibitor SC-514 was purchased from Cayman Chemical. Antibodies used in this study are shown in Supplementary Table 1.

### Isolation and culture of primary fibroblasts
Primary fibroblasts were obtained from wild-type and *C15orf48*[−/−] littermate embryos (E14.5). The embryos' head and visceral tissues were

removed, and the remaining tissues were minced with cell scrapers. Subsequently, the tissue fragments were incubated in a solution containing 0.25% trypsin and 1 mM EDTA (Fujifilm Wako) for 15 min at 37 °C. After incubation, the suspension was then filtered through a 100-μm nylon mesh and centrifuged at $180 \times g$ for 5 min. The cell pellet was resuspended and cultured in DMEM containing 10% FBS at 37 °C. After 4–6 h, nonadherent cells were removed, and adherent fibroblasts were cultured until confluence was achieved. The medium was replaced every other days, and cells were used at passages 3 to 5.

## Western blotting
Cells were washed with phosphate-buffered saline (PBS), directly lysed with Laemmli buffer, and used as whole-cell lysates. Testes from 4-week-old male mice and tissues from $Rag1^{-/-}$ female mice were washed with PBS and then homogenized in TNE buffer (10 mM Tris-HCl, 150 mM NaCl, 10 mM Triton X-100, and cOmplete (Sigma-Aldrich); pH 7.4). Homogenized tissues were sonicated and then centrifuged at $10,000 \times g$ for 5 min at 4 °C. Supernatants were collected and mixed with Laemmli buffer after measuring the protein concentration using a TaKaRa BCA Protein Assay Kit (TaKaRa, T9300A). Samples were subjected to SDS-polyacrylamide gel electrophoresis and electrotransferred onto polyvinylidene difluoride membranes. Protein bands were treated with appropriate antibodies for detection and analyzed using a ChemiDoc XRS+ image analyzer (Bio-Rad). Band intensity was measured using Quantity One software (Bio-Rad). Quantitative ratios were calculated based on the data and are presented as relative values. Western blotting was done in triplicate using three independent samples, and statistical analyses were done based on the quantified band intensities. Statistical significance was calculated using two-tailed unpaired Student's $t$ test or two-way ANOVA followed by Tukey's multiple comparisons test ($n = 3$, biological replicates) (Supplementary Figs. 12–18). Uncropped images are shown in the Source Data file.

## Quantitative real-time PCR (qPCR) analysis
Total RNA was isolated from cells using RNAiso Plus reagent (TaKaRa), and cDNA was synthesized from 0.5 μg of each RNA preparation using the ReverTra Ace qPCR RT kit (TOYOBO, FSQ-101) according to the manufacturer's instructions. The following primers were used for PCR: human *GAPDH*, 5′-GGA GCG AGA TCC CTC CAA AAT-3′ (sense) and 5′-GGC TGT TGT CAT ACT TCT CAT GG-3′ (antisense); and human *C15ORF48*, 5′-AGG AAG GAA CTC ATT CCC TTG G-3′ (sense) and 5′-TTT TGA GGT ACA GTA GGG TCC A-3′ (antisense). After initial denaturation at 95 °C for 1 min, PCR was performed for 40 cycles (15 s at 95 °C and 45 s at 60 °C) using a Thunderbird SYBR Green Polymerase Kit (TOYOBO, QPS-101) and Eco Real-Time PCR System (Illumina).

cDNAs from primary TEC subfractions were amplified by the RamDa method as described previously[42,74], with some modifications. Primary TEC subfractions were sorted using an Aria flow cytometer and lysed with lysis buffer for TurboCapture mRNA kits (QIAGEN, 72251). Cell lysates were cleaned with RNAClean XP (Beckman Coulter, A63987) and then used for reverse transcription with the RamDa method. The following primers were used for PCR: mouse *36B4*, 5′-TCC AGG CTT TGG GCA TCA-3′ (sense) and 5′-CTT TAT CAG CTG CAC ATC ACT-3′ (antisense); and mouse *C15orf48*, 5′-ATC TTT CGC TTT GTA TGC GTT GA-3′ (sense) and 5′-GGC TTC CAT TGC TGG TTG ATG-3′ (antisense). After initial denaturation at 98 °C for 2 min, PCR was performed for 40 cycles (10 s at 98 °C, 10 s at 60 °C, and 30 s at 68 °C) using a KOD SYBR qPCR mix (TOYOBO, QKD-201) and CFX96 Touch Real-Time PCR Detection System (BIO-RAD).

## Chromatin immunoprecipitation (ChIP) assay
ChIP assays were performed as described previously[75]. Briefly, cells were suspended in PBS at $2 \times 10^6$ cells/mL and fixed with 1% formaldehyde for 10 min at 25 °C, after which fixation was halted by addition of 150 mM glycine. Fixed cells were washed with PBS and

lysed with 200 mL of lysis buffer (50 mM Tris-HCl, 10 mM EDTA, and 1% SDS; pH 7.4). Cell lysates were sonicated using a bath sonicator (Bioruptor) for three cycles (power setting: high; on, 30 s, and off, 1 min; and 4 °C). Sonicated lysates were centrifuged for 10 min at $10,000 \times g$, and 1,800 mL of dilution buffer (50 mM Tris-HCl, 167 mM NaCl, 1.1% Triton X-100, and 0.11% sodium deoxycholate; pH 8.0) was added to the supernatants (chromatin solutions). The chromatin solution (2 mL) was pre-cleared with 80 μL of 50% protein G-Sepharose slurry pre-absorbed with 100 μg/mL sonicated salmon sperm DNA (ssDNA), aliquoted, and incubated with 4 μL of anti-RelA antibody (Santa Cruz) or 4 μg of normal mouse IgG (MOPC21; Merck Millipore) overnight. Immunoprecipitates were recovered using 20 μL of 50% protein G-Sepharose/ssDNA for 2 h, washed sequentially with RIPA buffer (50 mM Tris-HCl, 150 mM NaCl, 1 mM EDTA, 1% Triton X-100, and 0.1% SDS; pH 8.0), RIPA buffer with 500 mM NaCl, and TE buffer (10 mM Tris-HCl and 1 mM EDTA; pH 8.0), and resuspended in 200 μL of elution buffer (10 mM Tris-HCl, 300 mM NaCl, 5 mM EDTA, and 0.5% SDS; pH 8.0). Beads and an input fraction saved before pre-clearing were incubated at 65 °C for at least 6 h. DNA was extracted with phenol-chloroform, precipitated with ethanol, and suspended in 50 μL of TE buffer. Extracted DNA was subjected to qPCR analysis as described above. The following PCR primers (between −307 and −100 from the transcription start site of the *C15ORF48* gene predicted from RefSeq NM_032413.4) were used: 5′-GTT CCA CCT CCT ACT CCC CA-3′ (sense) and 5′-GAG GGA CCT GAC TCG CTT TC-3′ (antisense). Percent input values were calculated by comparing $Ct$ values of the input and immunoprecipitated fractions and are shown as ratios relative to those of the control samples.

## Mitochondrial membrane potential determination
Mitochondrial membrane potential was determined using tetramethylrhodamine methyl ester (TMRM) (Invitrogen). Cells were plated in 12-well plates and transfected with the indicated siRNAs for the indicated times. After incubation, cells were incubated with 30 nM TMRM for 30 min at 37 °C. After incubation, cells were collected in 1.5 mL tubes and washed with PBS three times. The median fluorescence intensity of TMRM was measured using an Aria flow cytometer (BD Biosciences). Gating strategy is shown in Supplementary Fig. 19a.

## Mitophagy detection
Mitophagy was detected using a Mitophagy Detection Kit (DOJINDO, #MD01), according to the manufacturer's instructions. Briefly, cells were seeded in a 12-well plate and incubated for 24 h. After incubation, the culture medium was discarded, and cells were washed twice with serum-free medium. Cells were then treated with mitophagy dye working solution and incubated at 37 °C for 30 min. To induce mitophagy, cells were additionally treated with the mitochondrial uncoupler CCCP (50 μM) for 2 h after treatment with the Mitophagy dye working solution. After incubation, supernatants were discarded, and cells were washed twice with serum-free medium and collected in a 1.5 mL tube. Mitophagy-dye-positive cells were detected using an Aria flow cytometer (BD Biosciences). CCCP-treated cells (CCCP+) were analyzed for positive control of mitophagy. Gating strategy is shown in Supplementary Fig. 19b.

## ATP assay
The ATP assay was performed using an ATP assay kit-Luminescence (DOJINDO, #A550) according to the manufacturer's instructions. Briefly, cells were transfected with the indicated siRNAs using the reverse transfection method with Lipofectamine RNAiMax and stimulated with IL-1α for the indicated time periods. After incubation, cells were detached and suspended in the culture medium at 2,000,000 cells/mL. Cell suspensions (200,000 cells/100 μL) were plated in 96-well, white microplates and incubated for 4 h. ATP standard solutions were prepared by serial dilution with serum-free

medium at 2.5, 1.25, 0.625, 0.313, 0.156, 0.078, 0.039, and 0 μM. Each diluted ATP standard solution was added to a 96-well, white microplate. After incubation, working solution was added to each well. The microplate was covered with foil, shaken for 2 min, and then incubated at 25 °C for 10 min. Luminescence was measured using a Filter Max F5 microplate reader (Molecular Devices). ATP concentrations of cell suspensions (200,000 cells/100 μL) were calculated using a calibration curve, and average ATP amount in a cell was determined.

### Glutathione assay
Glutathione assays were performed using a GSSG/GSH Quantification kit (DOJINDO, #G257) according to the manufacturer's instructions. Briefly, cells were plated in a 6-cm dish and transfected with the indicated siRNAs using the reverse transfection method with Lipofectamine RNAiMax for the indicated times. After incubation, cells were collected and adjusted to $1 \times 10^7$ cells/1.5 mL tube. Cell suspensions were mixed with 10 mM HCl and then lysed with two freeze-thaw cycles (30 min at −80 °C and 30 min at 4 °C). Five percent 5-Sulfosalicylic Acid Dihydrate (SAA) was added to samples, which were then centrifuged at $10,000 \times g$ for 10 min at 4 °C. Supernatants were used for GSSG/GSH quantification. Standard solutions of GSSG and GSH were prepared by serial dilution with 0.5% SAA. Each diluted standard solution was then added to wells of a 96-well microplate. Buffer solution was added to each well and incubated at 37 °C for 1 h. Substrate working solution and enzyme/coenzyme working solution were added to each well, and then the plate was incubated at 37 °C for 10 min. Absorbance was measured at 405 nm using a Filter Max F5 microplate reader (Molecular Devices). Cellular GSSG and GSH concentrations were calculated using a calibration curve.

### MTT (3-[4,5-dimethylthiazol-2-yl]−2,5 diphenyl tetrazolium bromide) assay
Cells were plated in 96-well microplates and transfected with the indicated siRNAs using the reverse transfection method with Lipofectamine RNAiMax for the indicated time periods. After incubation, the solution in each well was replaced with DMEM containing 0.5 mg/mL MTT (DOJINDO) and incubated for an additional 1 h. The resulting crystalized product was dissolved in 100 μL of 100% dimethyl sulfoxide (DMSO) and absorbance was measured at 595 nm using a Filter Max F5 microplate reader (Molecular Devices).

### Immunocytochemistry
Images were obtained using an LSM 780 confocal laser scanning microscope with a $63 \times 1.40$ NA oil-immersion objective (Zeiss). A single planar (xy) slice (1.0-μm thickness) is shown in all experiments. Cells were fixed in 4% paraformaldehyde for 20 min at room temperature and permeabilized in PBS containing 0.2% Triton X-100 and 3% bovine serum albumin at room temperature. Cells were subsequently reacted with anti-C15ORF48/NMES1 (NBP1-98391) antibody for 24 h or anti-LC3 (MBL, PM036) antibody for 1 h, washed with PBS containing 0.1% saponin, and stained with an Alexa Fluor 488- or 594-secondary antibody for 1 h. Nuclei were stained with a mixture of 20 μg/mL propidium iodide (PI) or 4′,6-diamidino-2-phenylindole (DAPI) and 200 μg/mL RNase A for 30 min. Mitochondria was stained with 100 nM Mitotracker CMXRos (Thermofisher Scientific). Stained cells were mounted with ProLong Antifade reagent (Thermo Fisher Scientific).

### Preparation of TEC suspensions and flow cytometry
Mouse thymi were minced with razor blades. Thymic fragments were then pipetted up and down to remove lymphocytes. Fragments were digested with RPMI 1640 medium containing Liberase (Roche, 0.05 U/mL) and DNase I (Sigma-Aldrich) and incubated at 37 °C for 12 min three times. Single-cell suspensions were then stained with anti-mouse antibodies. Dead cells were excluded via 7-aminoactinomycin

D (7-AAD) or SYTOX Blue staining. For staining TECs with the C15ORF48 antibody (Novus Biologicals) and the analysis of Tregs, TEC suspensions and thymocytes were fixed and permeabilized with Foxp3 staining buffer set (eBioscience, 00-5523-00) according to the manufacturer's instructions and used for intracellular staining. Cells were sorted using an Aria flow cytometer (BD Biosciences). Data were analyzed using FlowJo 10 software. Gating strategies are shown in Supplementary Figs. 20–22.

### GFP-LC3 mice analysis
For autophagy analysis of thigh skeletal muscle, $C15orf48^{-/-}$/GFP-LC3 mice and GFP-LC3 control mice were maintained without food for 48 h, but with free access to drinking water. Before analysis, mice were anaesthetized and perfused through the left ventricle with 4% paraformaldehyde in PBS. Thymi were collected from normally fed mice and fixed with 4% paraformaldehyde in PBS for 4 h, followed by treatment with 15% sucrose in PBS 4 h at room temperature, and 30% sucrose in PBS overnight at 4 °C. Thymic samples were embedded in optimal cutting temperature (OCT) compound and stored at −80 °C. Sections (5-μm thickness) of the thymus in OCT compound were mounted on glass slides coated with aminosilane and fixed with ice-cold acetone for 5 min. After washing, sections were blocked with 10% goat serum and then stained with Keratin 5 or Keratin 8 together with GFP antibody (Abcam) in 10% goat serum for overnight at 4 °C. After washing and further incubation with fluorescence-labeled secondary antibodies for 1 h, sections were covered with glass coverslips. Images were acquired using a BZ-X710 microscope (Keyence). Areas of keratin 5- or 8-positive regions and muscle fibers and the number of LC3 puncta were calculated using BZ-X710 software (Keyence).

### Analysis of scRNA-seq
The DDBJ database of mouse thymic epithelial cells (DRA009125) [https://ddbj.nig.ac.jp/resource/sra-submission/DRA009056] was used for this analysis. Plots were created using Seurat v4.1.1 package. Cell types were defined based on expression markers as described previously[42].

### Immunohistochemistry
Tissues from $Rag1^{-/-}$ female mice were embedded in OCT compound (Sakura) and frozen in liquid nitrogen. Cryostat sections (5 μm) were fixed with ice-cold acetone for 5 min. Tissue sections were blocked with 10% goat serum and then incubated with sera from 21-week-old wild-type and $C15orf48^{-/-}$ mice (100× dilution) for 1 h at room temperature. Slides were subsequently incubated with a secondary antibody (anti-mouse IgG-Alexa Fluor-488) and DAPI for 40 min at room temperature. Images were obtained using a confocal laser-scanning microscope (Leica). For the analysis of glomerulitis, kidney sections from of 21-week-old wild-type and $C15orf48^{-/-}$ littermate mice were prepared as described above and then stained with anti-mouse IgG-Alexa Fluor-488 and DAPI for 40 min at room temperature. IgG deposits were scored in a double-blinded manner on a scale of 0–4 (0 = negative, 1 = weak, 2 = moderate, 3 = strong, 4 = maximal fluorescence) in five different fields for each kidney section[76]. All images were acquired at the same exposure time to allow comparison among samples.

### Fetal thymic stroma transplantation
$C15orf48^{-/-}$ mice crossed with wild-type BALB/c mice more than 10 times were used for thymic transplantation. The isolated fetal thymic lobes (E15.5) were cultured on Nucleopore filters (Whatman) in R10 medium, which consists of RPMI 1640 (Fujifilm Wako), 10% fetal bovine serum (Equitech-Bio), 2 mM L-glutamine (Fujifilm Wako), 100 U/mL penicillin (Banyu Pharmaceutical), 100 μg/mL streptomycin (Meiji Seika Pharma Co., Ltd.), 50 μM 2-mercaptoethanol (Fujifilm Wako) and 1.35 mM 2′-deoxyguanosine (Sigma-Aldrich) for 4 days. The cultured

fetal thymus (fetal thymic stroma) underwent an additional day of culture in R10 medium without 2′-deoxyguanosine. Subsequently, these fetal thymic stromas were transplanted into the kidney capsules of 6-week female nude mice on the BALB/c background. Eight weeks later, sera and tissue sections from transplanted nude mice were analyzed for presence of autoimmune phenotypes.

### Detection of inflammatory cell infiltration
Mouse tissues were harvested and fixed overnight in 4% paraformaldehyde (Nacalai), embedded in paraffin, sectioned, and stained with hematoxylin and eosin. The degrees of inflammatory cell infiltration were scored in a double-blinded manner on a scale of 0–4 (0 = no detectable infiltration, 1 = a focus of perivascular infiltration, 2 = several foci of perivascular infiltration, 3 = cellular infiltration in >50% of the vasculature, 4 = severe infiltration throughout the interstitial region)[77].

### Quantification and statistical analysis
Graphs are presented as mean ± SD, as indicated in the figure legends. Three or more independent replicates were performed for each experiment. Two-tailed unpaired Student's $t$ test was used for comparison between two groups. Comparisons of more than two groups were performed using two-way ANOVA followed by Tukey's multiple comparison test. A $p$ value less than 0.05 was considered statistically significant. Exact $p$ values were rounded to the fifth decimal place and are shown in the graphs. $P$ values less than 0.001 or 0.0001 are shown as <0.001 or <0.0001. Sample sizes can be found in the main and supplementary figure legends.

### Reporting summary
Further information on research design is available in the Nature Portfolio Reporting Summary linked to this article.

## Data availability
All these data supporting the findings of this study are available in the paper and its supplementary information files or from the corresponding authors upon request. The DDBJ database of mouse thymic epithelial cells (DRA009125) [https://ddbj.nig.ac.jp/resource/sra-submission/DRA009056] was used for the scRNA-seq analysis. Source data are provided with this paper.

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

## Acknowledgements

We thank Dr. Noboru Mizushima for GFP-LC3 mice, Dr. Yasuyuki Kurihara, Dr. Kazumasa Aoyama, Mr. Takuro Araki, and Mr. Naoto Hori for their technical assistance, and members of the central facilities of RIKEN IMS for generation of *C15orf48*$^{-/-}$ mice. This work was supported in part by grants-in-aid for Challenging Research (Exploratory) [grant number 19K22482 (to N.Ya.)], Scientific Research (C) [grant numbers 21K08121 (to H.T.) and 21K06543 (to N.Ya.)], and Research Activity Start-up [grant number 22K20706 (to Y.T.)] from the Japanese Ministry of Education, Culture, Sports, Science, and Technology, Chiba Foundation for Health Promotion & Disease Prevention (to Y.T.), the Suzuken Memorial Foundation (to Y.T.), Takeda Science Foundation (to N.Ya.), TERUMO Life Science Foundation (to N.Ya.), and the Research Foundation for Pharmaceutical Sciences (to N.Ya.), Grants-in-Aid for Scientific Research from JSPS (20K07332, 20H03441) (T.A., N.A.), and CREST from Japan Science and Technology Agency (JPMJCR2011) (T.A. and T.J.K.).

## Author contributions

Y.T., T.A. and N. Ya. designed the study, performed experiments, analyzed results and wrote the manuscript. M.Ma., N.T., Y.K., S.K., K.H., M.Mi., T.I., N.A., T.S., T.M., M.H., R.E., H.I., Y.M., N.H., T.J.K., Na. Y., and H.T. assisted with experiments and provided several reagents. T.A. and N.Ya. supervised and directed the research. All authors discussed the results and commented on the manuscript.

## Competing interests

The authors declare no competing interests.
