## [Peer Review File · Nature Communications]

Mitochondrial protein C15ORF48 is a stress-independent inducer of autophagy that regulates oxidative stress and autoimmunityREVIEWER COMMENTS

Reviewer #1 (Remarks to the Author):

In this study, Takura et al investigate the regulatory role of the mitochondrial protein C15ORF48 in autophagy. The authors began by presenting convincing results, derived from in vitro experiments with human lung cancer cell lines, indicating the existence of a C15ORF48-dependent mechanistic axis coupling the control of mitochondrial membrane potential and ATP levels and the activation of AMPK and the ULK-dependent autophagy (FIGURES 1-5). First, they showed that forced expression of C15ORF48 activates AMPK and ULK1 (FIG1), and augments autophagy (FIG2). Induction of endogenous C15ORF48 by IL-1alpha replicates the observations obtained with forced expression of C15ORF48 (FIG3). C15ORF48 Knockdown in MDA-MB-231 breast cancer cells, which present elevated levels of C15ORF48, further validated that C15ORF48 activates stress-independent autophagy in tumor cells (FIG4). Moreover, the study provides interesting evidence for a regulatory loop whereby C15ORF48 upregulates in an ATG5- and ATG7-dependent antioxidants (GSH) to balance ROS levels (FIG5). Previous studies implicated that autophagy regulation in TECs is important for their role in T cell selection [1-3]. In the last part of the study, Takura et al investigate the role of C15ORF48 in autophagy in TECs and its consequence for T cell immune development and tolerance. Using available scRNA data sets and flow cytometry analysis, the expression of C15ORF48 was mapped to distinct TEC subsets, including cTEC, Aire+ and post-Aire cells (FIG6). Analysis of C15ORF48 KO mice showed that C15ORF48-deficiency reduced cTEC and mTEC populations, without causing a major alteration in the numbers and global (DN_DP_SP4/SP8) T cell developmental program (FIG6 and EXTENDED DATA FIG 3B). Analysis of SP4 maturation showed a moderate effect in M2 cells. Moreover, immunofluorescence analysis of thymic section showed reduced GFP-LC3 staining in TECs from C15ORF48 KO mice. Lastly, the authors showed evidence that the serum of 21wk old C15ORF48 KO mice presented a higher amount of autoantibodies (FIG7).

Although the study is interesting, and the observations within are intriguing, my main criticism pertains to the lack of mechanistic insights potentially explaining on how C15ORF48-deficiency in TECs may lead to failures in tolerance. There are a series of shortcomings in the analysis of autophagy in C15ORF48-deficient TECs and T cell development, which in the view of this reviewer would need to be addressed to strengthen the manuscript's conclusions. In the absence of clear insights linking the observations made in the thymus and their consequences in peripheral tolerance, the authors should attempt to provide more results on the analysis of the thymus and pull back on some conclusions based on the provided results.

1. Past, and more recent studies have suggested that autophagy regulation in TEC, namely cTECs, is important for their role in CD4 T cell selection [1-3]. The effect in M2 SP4 is low and the analysis of T cell development in the thymus of C15ORF48 KO mice is superficial. It is not clear if C15ORF48-deficiency affects positive, negative, or regulatory T cell selection in thymus. This can be investigated in the polyclonal setting (TCRbeta/CD69; PD1/Helios) or by crossing C15ORF48 KO onto TCR tg background.
2. This reviewer infers that C15ORF48 may be expressed in other cell types apart from TECs. As the analysis was conducted with germline C15ORF48 KO mice, the interpretation of thymus and peripheral (autoantibodies) phenotypes may be caused by confounding effects caused by C15ORF48 deficiency. In the absence of a cKO mouse model, if possible, the authors can consider performing thymic transplantation experiments of C15ORF48 KO thymic into athymic nude mice, thereby restricting C15ORF48-deficiency to the thymic stroma compartment. Alternatively this point/limitation should be discussed.
3. The results regarding the defects in autophagy in TEC are not very clear. What is the area of the thymus represented in FIG7A, the cortical or medullary region? It has been previously demonstrated that cTEC presented elevated levels of autophagy [1, 2]. Because C15ORF48 is also expressed in cTEC

(FIG6), autophagy should be measured in cTEC and mTEC from mutant thymus.

4. The authors may consider discussing their results in line with the findings described in two recent studies that autophagy regulation in TECs (cTEC) control CD4 T cell selection.

References:

1. Nedjic, J., et al., Autophagy in thymic epithelium shapes the T-cell repertoire and is essential for tolerance. *Nature*, 2008. 455(7211): p. 396-400.
2. Rodrigues, P.M., et al., LAMP2 regulates autophagy in the thymic epithelium and thymic stroma-dependent CD4 T cell development. *Autophagy*, 2022: p. 1-14.
3. Postoak, J.L., et al., Thymic epithelial cells require lipid kinase Vps34 for CD4 but not CD8 T cell selection. *J Exp Med*, 2022. 219(10).

Reviewer #2 (Remarks to the Author):

Diverse stimuli promote remodelling of the mitochondrial cytochrome c oxidase complex (CIV), involving exchange between members of the NDUFA4 family of proteins at the outer edge of the complex. This manuscript addresses the role of a newly-discovered member of the NDUFA4 family, encoded by the C15ORF48 gene in man (formally Nmes1 or AA467197 in mouse, but here referred to as C15ORF48). It is proposed that C15ORF48 reduces CIV activity, mitochondrial membrane potential and mtROS production, impairs ATP production and promotes autophagy, which in turn drives self-tolerance by assisting presentation of self antigens in thymic epithelial cells. The manuscript is heavily dependent on genetic manipulation of C15ORF48 levels by over-expression or siRNA-mediated knockdown in human airway epithelial or breast cancer cell lines. The last two of seven figures in the main body of the manuscript make use of a novel C15ORF48 knockout mouse that the authors generated. The knockout is shown to have fewer autophagic puncta in thymic epithelium, and an increase in auto-reactive antibodies. This is an interesting story that potentially sheds new light on biological functions of the enigmatic C15ORF48 gene. However there are several important issues that still need to be addressed.

Assertions about effects of C15ORF48 on mitochondrial function are supported by over-expression experiments in transformed cell lines, both here and in cited references. C15ORF48-mediated changes in mitochondrial ROS generation are not demonstrated, but inferred from redox markers. The first problem is that aberrant cellular metabolism is a hallmark of transformation, making it notoriously difficult to extrapolate from cancer cells to primary cells. Second, the assembly of mitochondrial electron transport chains and higher order complexes is tightly regulated to match mitochondrial metabolism and ROS generation to cellular demands and challenges. It is possible that artefacts could be caused by over-expression of a mitochondrial protein that locates at the outside of one electron transport chain (ETC) complex and is known to participate in interactions between ETCs. There is some corroboration from siRNA knockdown experiments, but again using a cancer cell line. In primary LPS-activated macrophages, endogenous NDUFA4 protein in CIV is replaced by C15ORF48 [27], but it was not formally demonstrated that this exchange causes any alteration of mitochondrial function. Overall, there is not yet compelling evidence as to what changes of mitochondrial function are driven by endogenous C15ORF48. The manuscript would be stronger if changes in mitochondrial function were demonstrated in primary C15ORF48^{-/-} cells. With the generation of a new knockout mouse strain the authors have a fantastic opportunity to do this, potentially linking mitochondrial function directly to an auto-immune phenotype. Overexpression of NDUFA4L2 or NDUFA4 might also be used to demonstrate that the reported metabolic changes are specific to C15ORF48, rather than an artefact caused by disruption of higher order ETC complexes.

The auto-immune phenotype of C15ORF48^{-/-} mice is presented in quite sparse detail. Fig. 6 describes

some quite subtle changes of thymic epithelial cell populations, which I am not well qualified to comment on. The reduction of LC3 puncta in knockout thymic epithelium in Fig. 7 looks convincing and is consistent with the preceding in vitro data. But if the knockout has a true auto-immune phenotype one would expect physiological consequences. None are described, and the nature of putative auto-antibodies is not defined.

The manuscript is not consistently clear about experimental replication: for example whether graphs display technical replicates or outcomes of independent experiments; what "n=" statements actually mean; whether western blots are representative of several repeats with similar outcome. This last point is critical because several important conclusions rest on differences of band intensities that are quite small, not particularly obvious to the naked eye, and not always consistent between experiments. Key conclusions would be more strongly supported by presenting mean values +/- error from several independent experimental repeats.

Queries:

It is not explained why the CCCP+/- method is used to investigate effects of C15ORF48 knockdown on mitochondrial membrane potential. I find these data difficult to interpret. Likewise, why are relative rather than absolute ATP levels measured?

What proportion of LC3 puncta in vitro and in vivo contain mitochondria? If much of the LC3 signal is related to mitophagy of defective mitochondria, can this explain differences in self-antigen presentation?

Please comment on the efficacy of C15ORF48 knockout in vivo. Supplemental Fig 2d suggests efficient knockout in testes, but flow data suggest incomplete knockout in the most highly expressing cTEC population (isotype control would help interpretation of these data).

Reviewer #3 (Remarks to the Author):

Takakura et al investigate the role of the mitochondrial protein C15ORF48 in autophagy. Their experiments lead them to conclude that C15ORF48 induces autophagy independent of starvation or mitochondrial stress. The authors hypothesize that C15ORF48 might be a possible regulator of self-antigen production in TECs, specifically because it has this nonconventional autophagy-inducing function. The results of their experiments are interpreted to support this hypothesis. The manuscript reports a series of interesting experiments, the majority of which focus on establishing the unique autophagy-inducing capacity of C15ORF48. A minor part is concerned with the role of C15ORF48 in TECs.

In general, the experiments are well designed, although some of the results require additional validation and/or more work to substantiate the claims.

Specifically, I have the following concerns:

(1) In Fig. 1, I am puzzled by the discrepancy of results obtained by immunofluorescence and Western blot. In panel a, no signal is obtained for C15ORF48 in control cells, yet a clear signal is seen in panel d. Please, explain.

(2) In Fig. 2, it would seem to me that the presence of C15ORF48 in cells treated with CCCP reduces the mitophagy index. Does this mean that C15ORF48 is protective? Please, comment.

(3) In Fig. 3, si compounds directed against C15ORF48 are used. How was their specificity tested?

(4) With respect to the analysis of TECs, the authors show that C15ORF48 is expressed (at the RNA level) in many TEC subtypes, including the most mature cTECs (cluster 10 in Fig. 6a). Yet, the authors do not show IHC analysis of K8+ cells in Fig. 7a. Why? Is there a difference between RNA and protein levels?

(5) The intracellular staining profiles in Fig. 6b are not convincing, nor are the differences between TEC subtypes shown in Fig. 6d. They should be backed up by RNAseq or qPCR data to verify the subtle differences, as the isolation and analysis of TEC subsets is notoriously difficult (see PMID 29298829).

(6) The analysis of potential autoantibodies (Fig. 7c) needs to be controlled for genetic background,

about which we are not informed. Are the wild-type and mutant mice isogenic? The differences are not significant, except for the lacrimal gland. Is glomerulonephritis or orchitis observed in the mutant mice?

Reply to the Reviewers

We would like to thank all the reviewers for their constructive criticisms on our work.

REVIEWER COMMENTS

Reviewer #1 (Remarks to the Author):

In this study, Takura et al investigate the regulatory role of the mitochondrial protein C15ORF48 in autophagy. The authors began by presenting convincing results, derived from in vitro experiments with human lung cancer cell lines, indicating the existence of a C15ORF48-dependent mechanistic axis coupling the control of mitochondrial membrane potential and ATP levels and the activation of AMPK and the ULK-dependent autophagy (FIGURES 1-5). First, they showed that forced expression of C15ORF48 activates APMK and ULK1 (FIG1), and augments autophagy(FIG2). Induction of endogenous C15ORF48 by IL-1alpha replicates the observations obtained with forced expression of C15ORF48 (FIG3). C15ORF48 Knockdown in MDA-MB-231 breast cancer cells, which present elevated levels of C15ORF48, further validated that C15ORF48 activates stress-independent autophagy in tumor cells (FIG4). Moreover, the study provides interesting evidence for a regulatory loop whereby C15ORF48 upregulates in an ATG5- and ATG7-dependent antioxidants (GSH) to balance ROS levels (FIG5). Previous studies implicated that autophagy regulation in TECs is important for their role in T cell selection [1-3]. In the last part of the study, Takura et al investigate the role of C15ORF48 in autophagy in TECs and its consequence for T cell immune development and tolerance. Using available scRNA data sets and flow cytometry analysis, the expression of C15ORF48 was mapped to distinct TEC subsets, including cTEC, Aire+ and post-Aire cells (FIG6). Analysis of C15ORF48 KO mice showed that C15ORF48-deficiency reduced cTEC and mTEC populations, without causing a major alteration in the numbers and global (DN_DP_SP4/SP8) T cell developmental program (FIG6 and EXTENDED DATA FIG 3B). Analysis of SP4 maturation showed a moderate effect in M2 cells. Moreover, immunofluorescence analysis of thymic section showed reduced GFP-LC3 staining in TECs from C15ORF48 KO mice. Lastly, the authors showed evidence that the serum of 21wk old C15ORF48 KO mice presented a higher

amount of autoantibodies FIG7).

Although the study is interesting, and the observations within are intriguing, my main criticism pertains to the lack of mechanistic insights potentially explaining on how C15ORF48-deficiency in TECs may lead to failures in tolerance. There are a series of shortcomings in the analysis of autophagy in C15ORF48-deficient TECs and T cell development, which in the view of this reviewer would need to be addressed to strengthen the manuscript's conclusions. In the absence of clear insights linking the observations made in the thymus and their consequences in peripheral tolerance, the authors should attempt to provide more results on the analysis of the thymus and pull back on some conclusions based on the provided results.

1. Past, and more recent studies have suggested that autophagy regulation in TEC, namely cTECs, is important for their role in CD4 T cell selection [1-3]. The effect in M2 SP4 is low and the analysis of T cell development in the thymus of C15ORF48 KO mice is superficial. It is not clear if C15ORF48-deficiency affects positive, negative, or regulatory T cell section in thymus. This can be investigated in the polyclonal setting (TCRbeta/CD69; PD1/Helios) or by crossing C15ORF48 KO onto TCR tg background.

> As the reviewer's comment, we addressed whether the C15ORF48 deficiency affects positive selection (CD3/CD69), negative selection (CD4SP/Helios), and Treg development (CD4SP/CD25/Foxp3) in the thymus (new data in Supplementary Fig. 8a-c). Unexpectedly, our results revealed that these selection processes remained largely unaffected in the absence of C15ORF48. Our interpretation of these findings suggests that the autophagy-dependent processing of self-antigens by C15ORF48 may play a role in shaping only a subset of the T-cell receptor (TCR) repertoires, rather than governing the entire TCR selection process. To explore this notion further, a comprehensive analysis of TCR repertoires in mature thymic T cells should be a subject for future research. Notably, a difference in the C15ORF48 dependency between cTECs and mTECs in autophagy induction was suggested (new data in Fig. 7a,b), which may also explain the limited impact observed in thymic positive selection by the C15ORF48

deficiency. We would like to add a discussion regarding these points (page 11, line 358-368).

2. This reviewer infers that C15ORF48 may be expressed in other cell types apart from TECs. As the analysis was conducted with germline C15ORF48 KO mice, the interpretation of thymus and peripheral (autoantibodies) phenotypes may be caused by confounding effects caused by C15ORF48 deficiency. In the absence of a cKO mouse model, if possible, the authors can consider performing thymic transplantation experiments of C15ORF48 KO thymic into athymic nude mice, thereby restricting C15ORF48-deficiency to the thymic stroma compartment. Alternatively this point/limitation should be discussed.

> We appreciated this reviewer's comments. Because the role of C15ORF48 in TEC-mediated thymocyte selections is the most important issue of this study, we did transplantation experiments of thymic stroma of *C15orf48*-deficient mice into athymic nude mice. As shown in new data in Fig. 9a-c, transplantation of *C15orf48*-deficient thymic stroma caused obvious autoimmune phenotypes in nude mice, such as increased autoantibodies and infiltration of inflammatory cells into various tissues. These results strengthen our conclusion that C15ORF48-deficiency in TECs causes autoimmunity.

3. The results regarding the defects in autophagy in TEC are not very clear. What is the area of the thymus represented in FIG7A, the cortical or medullary region? It has been previously demonstrated that cTEC presented elevated levels of autophagy [1, 2]. Because C15ORF48 is also expressed in cTEC (FIG6), autophagy should be measured in cTEC and mTEC from mutant thymus.

> We apologize for our insufficient explanation. In original Fig. 7a, we showed reduction in autophagy in mTEC regions in thymus by detecting a mTEC marker Keratin 5. During this revision, we obtained anti-Keratin 8 antibodies to detect cTEC regions and performed immunohistological analyses of thymic sections from *C15orf48*-deficient mice crossed with GFP-LC3 mice. As shown in New data in Fig. 7b, autophagy activity was slightly but significantly reduced in Keratin 8-positive cTECs in thymus of *C15orf48*-deficient mice, suggesting that C15ORF48 is also involved in

constitutive autophagy in cTECs. Importantly, reduction in autophagy in cTECs was weak as compared with that in mTECs in *C15orf48*-deficient mice, and C15ORF48 deficiency did not affect positive selection of thymocytes (new data in Supplementary Fig. 8a). Therefore, we hypothesize that C15ORF48 may partially contribute to autophagy-mediated presentation of self-antigens to thymocytes in cTECs. We added a discussion regarding these points (page 11, line 358-368).

4. The authors may consider discussing their results in line with the findings described in two recent studies that autophagy regulation in TECs (cTEC) control CD4 T cell selection.

References:

- 1. Nedjic, J., et al., Autophagy in thymic epithelium shapes the T-cell repertoire and is essential for tolerance. *Nature*, 2008. 455(7211): p. 396-400.**
- 2. Rodrigues, P.M., et al., LAMP2 regulates autophagy in the thymic epithelium and thymic stroma-dependent CD4 T cell development. *Autophagy*, 2022: p. 1-14.**
- 3. Postoak, J.L., et al., Thymic epithelial cells require lipid kinase Vps34 for CD4 but not CD8 T cell selection. *J Exp Med*, 2022. 219(10).**

> As mentioned in the reply to comment 3, we investigated autophagy activity in cTECs of *C15orf48*-deficient mice and showed that C15ORF48 is partially involved in constitutive autophagy in cTECs (New data in Fig. 7b). In addition, we analyzed positive selection of thymocytes by detecting expression of CD3 and CD69 and found that this population was largely unaffected in *C15orf48*-deficient mice (new data in Supplementary Fig. 8a). Therefore, although C15ORF48 has substantial involvement in autophagy in cTECs, its role in cTEC-mediated self-antigen presentation to thymocytes is limited. We added these discussion comments with citation of above-mentioned reference 2 and 3 (page 11, line 358-368).

Reviewer #2 (Remarks to the Author):

Diverse stimuli promote remodelling of the mitochondrial cytochrome c oxidase complex (CIV), involving exchange between members of the NDUFA4 family of proteins at the outer edge of the complex. This manuscript addresses the role of a

newly-discovered member of the NDUFA4 family, encoded by the C15ORF48 gene in man (formally Nmes1 or AA467197 in mouse, but here referred to as C15ORF48). It is proposed that C15ORF48 reduces CIV activity, mitochondrial membrane potential and mtROS production, impairs ATP production and promotes autophagy, which in turn drives self-tolerance by assisting presentation of self antigens in thymic epithelial cells. The manuscript is heavily dependent on genetic manipulation of C15ORF48 levels by over-expression or siRNA-mediated knockdown in human airway epithelial or breast cancer cell lines. The last two of seven figures in the main body of the manuscript make use of a novel C15ORF48 knockout mouse that the authors generated. The knockout is shown to have fewer autophagic puncta in thymic epithelium, and an increase in auto-reactive antibodies. This is an interesting story that potentially sheds new light on biological functions of the enigmatic C15ORF48 gene. However there are several important issues that still need to be addressed.

Assertions about effects of C15ORF48 on mitochondrial function are supported by over-expression experiments in transformed cell lines, both here and in cited references. C15ORF48-mediated changes in mitochondrial ROS generation are not demonstrated, but inferred from redox markers. The first problem is that aberrant cellular metabolism is a hallmark of transformation, making it notoriously difficult to extrapolate from cancer cells to primary cells. Second, the assembly of mitochondrial electron transport chains and higher order complexes is tightly regulated to match mitochondrial metabolism and ROS generation to cellular demands and challenges. It is possible that artefacts could be caused by over-expression of a mitochondrial protein that locates at the outside of one electron transport chain (ETC) complex and is known to participate in interactions between ETCs. There is some corroboration from siRNA knockdown experiments, but again using a cancer cell line. In primary LPS-activated macrophages, endogenous NDUFA4 protein in CIV is replaced by C15ORF48 [27], but it was not formally demonstrated that this exchange causes any alteration of mitochondrial function. Overall, there is not yet compelling evidence as to what changes of mitochondrial function are driven by endogenous C15ORF48. The manuscript would be stronger if changes in mitochondrial function were

demonstrated in primary C15ORF48^{-/-} cells. With the generation of a new knockout mouse strain the authors have a fantastic opportunity to do this, potentially linking mitochondrial function directly to an auto-immune phenotype.

> We appreciated this reviewer's comments. Experiments using primary thymic epithelium (TECs) are difficult because the specific properties of mouse TECs quickly change when they are isolated from the thymus. Therefore, we obtained primary fibroblasts from mouse embryos (E14.5) and used for analyses of mitochondrial membrane potential, ATP levels, basal autophagy, and glutathione levels those were examined in A549 human lung cancer cells. As shown in new data in Supplementary Fig. 5a-f, significant upregulation in mitochondrial membrane potential and ATP levels and downregulation in basal autophagy and glutathione levels were observed in *C15orf48*-deficient fibroblasts. These results are consistent with the data obtained from experiments with A549 cells and strengthen our conclusions that C15ORF48 lowers mitochondrial membrane potential and ATP levels and in turn increases autophagy activity and glutathione levels.

Overexpression of NDUFA4L2 or NDUFA4 might also be used to demonstrate that the reported metabolic changes are specific to C15ORF48, rather than an artefact caused by disruption of higher order ETC complexes.

> As this reviewer's comments, we established stable A549 lines expressing NDUFA4 or NDUFA4L2 and analyzed mitochondrial membrane potential, ATP levels, basal autophagy, and glutathione levels in these cells. As shown in new data in Supplementary Fig. 3a-g, no apparent difference was detected in these cells about the examined phenotypes. These results suggest that C15ORF48 has an unique role in controlling mitochondrial membrane potential, ATP levels, basal autophagy, and glutathione levels among the NDUFA4 family.

The auto-immune phenotype of C15ORF48^{-/-} mice is presented in quite sparse detail. Fig. 6 describes some quite subtle changes of thymic epithelial cell populations, which I am not well qualified to comment on. The reduction of LC3 puncta in knockout thymic epithelium in Fig. 7 looks convincing and is consistent

with the preceding in vitro data. But if the knockout has a true auto-immune phenotype one would expect physiological consequences. None are described, and the nature of putative auto-antibodies is not defined.

> We appreciated and agreed with this reviewer's comments. We performed additional experiments to evaluate autoimmune phenotypes in *C15orf48*-deficient mice. Western blotting using sera from the mutant mice showed higher immunoreactivities than those from control mice against lung, kidney, and liver lysates from *Rag1*-deficient mice (lacking endogenous IgGs), indicating that induction of autoantibodies in *C15orf48*-deficient mice (New data in Fig. 8b and Supplementary Fig. 11a). Furthermore, increased infiltration of inflammatory cells were also observed in lung, kidney, and liver sections from *C15orf48*-deficient mice (new data in Fig. 8c). Upregulation in deposits of IgG was also detected in kidney sections from *C15orf48*-deficient mice, suggesting that the mutant mice have glomerulitis-like phenotypes (new data in Fig. 8b). These results strengthen our conclusion that *C15orf48*-deficient mice have autoimmune phenotypes.

The manuscript is not consistently clear about experimental replication: for example whether graphs display technical replicates or outcomes of independent experiments; what “n=” statements actually mean;

> We apologize for our insufficient explanation. Most of the experiments were done more than three times with independent biological samples. In these cases, “n” means the number of independent experiments with independent biological samples, and we added the term “biological replicates” in figure legends. In immunocytochemistry of LC3 (New Fig. 2c, 2f, 3f, 3i, 3l, 4c, 4g, Supplementary Fig. 1d, 3f, and 5d) and immunohistochemistry of GFP-LC3 (New Fig. 7a, b), “n” means the number of cells and sections analyzed in each experiment, respectively. We added this explanation in each figure legend.

whether western blots are representative of several repeats with similar outcome. This last point is critical because several important conclusions rest on differences of band intensities that are quite small, not particularly obvious to the naked eye,

and not always consistent between experiments. Key conclusions would be more strongly supported by presenting mean values +/- error from several independent experimental repeats.

> As this reviewer's comments, we conducted western blotting as three independent experiments using independent biological samples. We also performed quantitation of band intensities and statistical analyses. Representative data are shown in main figures, and replications and graphs are shown in New Supplementary Fig. 12-18. These data support reproducibility of our western blots.

Queries:

It is not explained why the CCCP+/- method is used to investigate effects of C15ORF48 knockdown on mitochondrial membrane potential. I find these data difficult to interpret.

> We apologize for inappropriate data preparation. According to the related study (Lee CQE et al. Nat Commun, 2021), we measured median TMRM fluorescence intensities and showed these values as mitochondrial membrane potentials (new data in Fig. 1b, 3b, Supplementary Fig. 3c, and 5b). These new data are consistent with our original conclusion that C15ORF48 suppresses mitochondrial membrane potential.

Likewise, why are relative rather than absolute ATP levels measured?

> We apologize for inconvenience of our data preparation about ATP levels. As described in the Method section, we measured an ATP concentration of cell suspension containing 200,000 cells and calculated an average ATP amount of a cell. We showed an ATP amount of a cell as a cellular ATP level in the revised manuscript (new 1c, 3c, 4d, Supplementary Fig. 3d, 5c).

What proportion of LC3 puncta in vitro and in vivo contain mitochondria? If much of the LC3 signal is related to mitophagy of defective mitochondria, can this explain differences in self-antigen presentation?

> As this reviewer's comments, we analyzed colocalization of LC3 puncta with mitochondria under the normal culture condition and found that there is no difference in their colocalization levels between control and C15ORF48-expressing A549 cells (new data in Supplementary Fig. 1c, d). We further investigated the relationship between C15ORF48 and mitophagy using the mitochondrial uncoupler CCCP and found that C15ORF48 has a suppressive, rather than promotive, role in CCCP-induced mitophagy. These additional data strongly support our original conclusion that C15ORF48-dependent autophagy is different from mitophagy.

We also tried to analyze colocalization of LC3 and mitochondria in vivo using thymic sections from GFP-LC3 mice. However, as observed in data in Fig. 7a,b, TECs are compacted and have little cytosol, and therefore it is difficult to evaluate colocalization of LC3 and mitochondria in a thymic cell using thymic sections. Because our in vitro experiments with A549 cells demonstrate that C15ORF48 does not induce mitophagy, we believe that phenotypes in *C15orf48*-deficient mice is independent of mitophagy.

Please comment on the efficacy of C15ORF48 knockout in vivo. Supplemental Fig 2d suggests efficient knockout in testes, but flow data suggest incomplete knockout in the most highly expressing cTEC population (isotype control would help interpretation of these data).

> We established *C15orf48*-deficient mice using CRISPR-Cas9 system with mouse fertilized eggs and confirmed homozygous deletion of *C15orf48* genes (data in Supplementary Fig. 4c, d). In the analysis of protein expression in TECs, it is hard to conduct western blotting owing to the limited cell numbers of these populations. Therefore, we did flow cytometry to examine C15ORF48 protein expression in TECs. As this reviewer's comments, anti-C15ORF48 antibodies showed high background signals in TECs, and it is difficult to evaluate C15ORF48 protein-dependent signals (difference between control TECs and *C15orf48*-deficient TECs). Thus, we did flow cytometry using isotype controls. As shown in New Fig. 6 b-e, the intensities of anti-C15ORF48 antibodies in *C15orf48*-deficient TECs are similar to those of isotype controls. In contrast, anti-C15ORF48 antibodies showed higher intensities in wild-type mTEC-hi, cTEC, Early-Aire mTEC, Late-Aire mTEC, and Post-Aire mTEC, suggesting significant C15ORF48 protein expression in these populations.

Reviewer #3 (Remarks to the Author):

Takakura et al investigate the role of the mitochondrial protein C15ORF48 in autophagy. Their experiments lead them to conclude that C15ORF48 induces autophagy independent of starvation or mitochondrial stress. The authors hypothesize that C15ORF48 might be a possible regulator of self-antigen production in TECs, specifically because it has this nonconventional autophagy-inducing function. The results of their experiments are interpreted to support this hypothesis.

The manuscript reports a series of interesting experiments, the majority of which focus on establishing the unique autophagy-inducing capacity of C15ORF48. A minor part is concerned with the role of C15ORF48 in TECs.

In general, the experiments are well designed, although some of the results require additional validation and/or more work to substantiate the claims.

Specifically, I have the following concerns:

(1) In Fig. 1, I am puzzled by the discrepancy of results obtained by immunofluorescence and Western blot. In panel a, no signal is obtained for C15ORF48 in control cells, yet a clear signal is seen in panel d. Please, explain.

> We apologize for our insufficient explanation. This discrepancy could be explained by difference in sensitivities of antibodies between immunofluorescence and western blotting. In general, antibodies show higher sensitivities in western blotting than in immunofluorescence because secondary antibodies used for western blotting are conjugated with enzymes (peroxidases) that powerfully enhance luminescence signals. In contrast, secondary antibodies for immunofluorescence are conjugated with fluorescent compounds. Because signal intensities in chemical luminescence are higher than those in fluorescence, western blotting could detect weak C15ORF48 protein expression in control cells (Fig. 1d).

(2) In Fig. 2, it would seem to me that the presence of C15ORF48 in cells treated with CCCP reduces the mitophagy index. Does this mean that C15ORF48 is protective? Please, comment.

> As this reviewer's comments, we analyzed the relationship between C15ORF48 and CCCP-induced mitophagy by detecting mitophagy dye-positive cells, accumulation of PINK1 proteins, and colocalization of LC3 puncta with mitochondria in CCCP-treated cells. As shown in new data in Supplementary Fig. 1a-d, CCCP-induced upregulation in these phenomena was significantly repressed in C15ORF48-expressing cells, suggesting that C15ORF48 has a suppressive role in mitophagy.

(3) In Fig. 3, si compounds directed against C15ORF48 are used. How was their specificity tested?

> We apologize for our insufficient explanation. We used two different siRNAs for C15ORF48 to confirm the specificity of these siRNAs. We could obtain similar results in knockdown experiments with these siRNAs (new Fig. 3a-l, 4d-g, 5g-i, and 5l-o), indicating that the effects of these siRNAs are derived from specific knockdown of C15ORF48 but not from off-target effects.

(4) With respect to the analysis of TECs, the authors show that C15ORF48 is expressed (at the RNA level) in many TEC subtypes, including the most mature cTECs (cluster 10 in Fig. 6a). Yet, the authors do not show IHC analysis of K8+ cells in Fig. 7a. Why? Is there a difference between RNA and protein levels?

> We appreciate this reviewer's comments. At the initial submission, we could not analyze autophagy activities in cTECs because of a lack of anti-Keratin 8 antibodies available for IHC. During this revision, we newly obtained anti-Keratin 8 antibodies that can be used for IHC and examined autophagy activities in cTECs. As shown in new data in Fig. 7b, *C15orf48*-deficiency significantly reduced basal autophagy activities in cTEC regions, indicating that C15ORF48 is involved in constitutive autophagy in cTECs.

(5) The intracellular staining profiles in Fig. 6b are not convincing, nor are the differences between TEC subtypes shown in Fig. 6d. They should be backed up by

RNaseq or qPCR data to verify the subtle differences, as the isolation and analysis of TEC subsets is notoriously difficult (see PMID 29298829).

> As this reviewer's comments, we examined *C15orf48* mRNA expression in mTECs and cTECs by qPCR analysis after sorting by flow cytometry. As shown in new data in Supplementary Fig. 6, significant expression of *C15orf48* mRNA was detected in mTEC-hi and cTEC subfractions but not in mTEC-lo. These data are consistent with those of intracellular staining of C15ORF48 proteins (Fig. 6b, c).

(6) The analysis of potential autoantibodies (Fig. 7c) needs to be controlled for genetic background, about which we are not informed. Are the wild-type and mutant mice isogenic?

> We apologize for our insufficient explanation. *C15orf48*-deficient mice and *Rag1*-deficiency mice used in Fig. 8a-d are both C57BL/6 background, and this information is described in the Method section. To highlight this important information, we added this information in the figure legend of revised manuscript. During this revision, we transplanted thymi from Balb/c background *C15orf48*-deficient mice into Balb/c nude mice (new data in Fig. 9a-c). This information is described in the Method section.

The differences are not significant, except for the lacrimal gland. Is glomerulonephritis or orchitis observed in the mutant mice?

> We appreciate this reviewer's comments and further investigated autoimmune phenotypes in *C15orf48*-deficient mice. Western blotting showed that sera from these mutant mice have higher immunoreactivities against tissue lysates as well as tissue sections from *Rag1*-deficient mice, indicating increased autoantibodies in *C15orf48*-deficient mice (New Fig. 8b). We also analyzed tissue sections from *C15orf48*-deficient mice and found increased infiltration of inflammatory cells in lung, kidney, and liver of these mutant mice (New Fig. 8c). Importantly, increased deposits of IgG was detected in glomerulus-like structures in *C15orf48*-deficient kidney sections, suggesting that these mutant mice have a glomerulitis phenotype (New data in Fig. 8d). These results strengthen our conclusion that *C15orf48*-deficient mice have autoimmune phenotypes.

REVIEWERS' COMMENTS

Reviewer #1 (Remarks to the Author):

The authors have addressed the concerns raised during the initial review, adding new results and discussing the different points, which partially improved the manuscript.

The inclusion of additional data, particularly the insights derived from thymic transplantations, contributes to supporting the identified defect in T cell selection arises for defects in thymic stroma. However, it is important to note that while positive (CD3/CD69), negative and T regulatory (Foxp3+) selection have been examined in a polyclonal setting of C15ORF48KO thymus (Supplemental Figure 8), but defects in these stages can not be ruled out. Essential data obtained from TCR Tg, TCR sequencing, and T regulatory suppressive function assays could have been attempted to enhance the robustness of the and potentially explain the underlying causes for thymic and autoimmune phenotype.

Furthermore, the revised manuscript does not provide conclusive mechanistic insights into the role of C15ORF48 in autophagy and TECs. Considering previous literature indicating differential autophagy rates between cTECs and mTECs in wild-type mice (Nedjic, J., et al., Nature 2008; Rodrigues et al Autophagy 2022), a more detailed examination of the presented in revised Figure 7 is warranted. The current data does not elucidate differences between cTEC and mTECs, despite the observed reduction in LC3 puncta in mutant TECs.

Clarification on how deficiency in C15ORF48KO potential contribute to alterations in autophagy, and how these in turn impact the function of TECs in T-cell selection is crucial for a comprehensive understanding of the reported findings.

Reviewer #2 (Remarks to the Author):

The authors have provided detailed answers to my questions and added substantial new data.

Reviewer #3 (Remarks to the Author):

In the revised manuscript, the authors have addressed my concerns. I have no further comments.

Reply to the Reviewers

We would like to thank all the reviewers for their constructive criticisms on our work.

REVIEWER COMMENTS

Reviewer #1 (Remarks to the Author):

The authors have addressed the concerns raised during the initial review, adding new results and discussing the different points, which partially improved the manuscript.

The inclusion of additional data, particularly the insights derived from thymic transplantations, contributes to supporting the identified defect in T cell selection arises for defects in thymic stroma. However, it is important to note that while positive (CD3/CD69), negative and T regulatory (Foxp3+) selection have been examined in a polyclonal setting of C15ORF48KO thymus (Supplemental Figure 8), but defects in these stages can not be ruled out. Essential data obtained from TCR Tg, TCR sequencing, and T regulatory suppressive function assays could have been attempted to enhance the robustness of the and potentially explain the underlying causes for thymic and autoimmune phenotype.

> We acknowledge the constructive suggestion to explore the impact on negative and positive selection through experiments using several TCR-Tg mice. However, it is worth noting that such experiments may pose challenges, necessitating the establishment of new mouse lines crossed with TCR-Tg mice, which is currently not immediately feasible. Similarly, TCR sequencing could offer valuable insights. Unfortunately, due to the observed lack of influence on either positive or negative selection in polyclonal TCR studies, conducting these experiments on a substantial number of mutant mice and their littermate control would be required to obtain reliable repertoire data. Given the significant time and cost implications associated with these implementations, we consider this aspect as a valuable avenue for future validation. In light of these constraints, we have emphasized the importance of future TCR repertoire analysis in our discussion and have also describe the possibility of TCR-Tg experiments (page 11, line 364-365).

Furthermore, the revised manuscript does not provide conclusive mechanistic

insights into the role of C15ORF48 in autophagy and TECs. Considering previous literature indicating differential autophagy rates between cTECs and mTECs in wild-type mice (Nedjic, J., et al., Nature 2008; Rodrigues et al Autophagy 2022), a more detailed examination of the presented in revised Figure 7 is warranted. The current data does not elucidate differences between cTEC and mTECs, despite the observed reduction in LC3 puncta in mutant TECs.

Clarification on how deficiency in C15ORF48KO potential contribute to alterations in autophagy, and how these in turn impact the function of TECs in T-cell selection is crucial for a comprehensive understanding of the reported findings.

> It is indeed intriguing to explore how the regulation of autophagy by C15ORF48 may differ between mTECs and cTECs. To comprehensively elucidate these differences, thorough biochemical studies investigating the effect of the C15ORF48 deletion on autophagy mechanisms in both mTECs and cTECs would be essential. Unfortunately, considering the very low number of TECs prepared from the thymus, obtaining immediate experimental results would be challenging. We acknowledge the significance of this aspect and recognize it as a valuable avenue for future research.

Reviewer #2 (Remarks to the Author):

The authors have provided detailed answers to my questions and added substantial new data.

> Thank you for reviewing our paper.

Reviewer #3 (Remarks to the Author):

In the revised manuscript, the authors have addressed my concerns. I have no further comments.

> Thank you for reviewing our paper.